# Normative modelling of molecular-based functional circuits captures clinical heterogeneity transdiagnostically in psychiatric patients
Timothy Lawn [1] ✉, Alessio Giacomel [1], Daniel Martins [1,2], Mattia Veronese [1,3], Matthew Howard [1], Federico E. Turkheimer [1] & Ottavia Dipasquale [1,4] ✉

Advanced methods such as REACT have allowed the integration of fMRI with the brain's receptor landscape, providing novel insights transcending the multiscale organisation of the brain. Similarly, normative modelling has allowed translational neuroscience to move beyond group-average differences and characterise deviations from health at an individual level. Here, we bring these methods together for the first time. We used REACT to create functional networks enriched with the main modulatory, inhibitory, and excitatory neurotransmitter systems and generated normative models of these networks to capture functional connectivity deviations in patients with schizophrenia, bipolar disorder (BPD), and ADHD. Substantial overlap was seen in symptomatology and deviations from normality across groups, but these could be mapped into a common space linking constellations of symptoms through to underlying neurobiology transdiagnostically. This work provides impetus for developing novel biomarkers that characterise molecular- and systems-level dysfunction at the individual level, facilitating the transition towards mechanistically targeted treatments.

Neuropsychiatric disorders present a formidable healthcare challenge for which we remain largely bereft of meaningful treatments. Indeed, first-line pharmacotherapies largely engage poorly understood pharmacological mechanisms discovered serendipitously decades ago[1] and remain ineffectual for many patients[2–4]. The reasons for this ultimately reflect the extreme difficulty of understanding how complex aberrations of cognition and affect map onto underlying neurobiology. This in turn relates to the overarching challenge of understanding the brain, which is best described as a complex system whose diverse constituent components interact across spatial and temporal scales to bring to bear the genetic and environmental interactions which collectively shape our experience and behaviour[5].

Neuropsychiatric disorders layer additional heterogeneous pathophysiology on top of this inherent neurobiological complexity, resulting in significant variability in both neural mechanisms and symptomatology which show similarities across, and differences within, diagnostic boundaries[6–11]. For this reason, many treatments are utilised trans-nosologically, with selective serotonin reuptake inhibitors (SSRIs) having FDA approval for the treatment of at least 10 different psychiatric disorders[12].

In short, the current diagnostic paradigm is undermined by symptoms and treatments that are non-specific within and across disorders and that are poorly mapped onto underlying neurobiology[13]. Despite this, the majority of research continues to largely employ a case-control-based paradigm, in which group average differences are characterised between clinical cohorts and matched healthy participants, inherently neglecting such heterogeneity. Precision psychiatry, and precision medicine more broadly, aims to move beyond large and poorly defined groups towards refined stratified or even individualised treatment based on underlying pathophysiological mechanisms[14]. For example, the Research Domain Criteria (RDoC) framework aims to re-examine mental disorders from the perspective of neurobehavioral functioning, regardless of conventional diagnostic categories[15–17]. This is nicely exemplified by a null result for the primary outcome measure of a clinical trial. Currently, this reflects the average treatment response across all patients included in the study. However, the substantial heterogeneity within diagnostic groups may mean that in fact, some patients do derive meaningful benefit (so-called "responders"), whilst others show no real improvement (so-called "non-responders"). Thus, the

[1]Department of Neuroimaging, Institute of Psychiatry, Psychology and Neuroscience, King's College London, London, UK. [2]Division of Adult Psychiatry, Department of Psychiatry, Geneva University Hospitals, Geneva, Switzerland. [3]Department of Information Engineering, University of Padua, Padua, Italy. [4]Department of Research & Development Advanced Applications, Olea Medical, La Ciotat, France. ✉e-mail: timothy.lawn@kcl.ac.uk; ottavia.dipasquale@olea-medical.com

failure of the trial to demonstrate the clinical utility of a given treatment may be due to the inherent limitations of the intervention studied, but could also reflect an inability to effectively stratify and the need for mechanistic biomarkers to enhance or supplant conventional diagnostic criteria within clinical practice.

Neuroimaging offers a non-invasive set of methods which facilitates the study of brain structure and function, providing insights into the neurobiology underpinning psychiatric disorders. However, within conventional analytic frameworks, neuroimaging data is typically analysed for differences between patients and controls as well as for relationships between patients' brain features and clinical measures. The latter offers insights into how variability across patients' brains relates to the severity of symptomatology. However, the ability to utilise these relationships across subjects to target treatment as well as transfer these relationships to apply within additional patients from outside the original cohort remains limited. A set of emerging methods have been developed to address this specific issue. Normative modelling aims to robustly characterise what certain aspects of brain structure or function "should" look like across the spectrum of healthy ageing based on a set of predictor variables, typically demographics such as age and sex[18–22]. This has been analogised to "growth charts for the brain" as it follows the same logic as assessing if a child's body measurement (e.g., height) shows typical growth given their age and sex, providing a reference for what is typical considering the variability in growth patterns[23]. Similarly in neuroimaging, once a model has been trained to make predictions about a given brain feature (e.g., cortical thickness, brain volume), it can be utilised to examine whether an individual with a given diagnosis, or particular constellation of symptoms, falls within the expected range or shows significant abnormalities as compared to the normal population. Such abnormalities, or deviations, can be either positive or negative based on whether they are greater or lower than the expected range, respectively. To date, normative models of structural grey and/or white matter measures have been explored in schizophrenia[24–27], bipolar disorder[25,27], Alzheimer's disease[28,29], ADHD[25,30], autism spectrum disorder[25,31,32], depression[25], and obsessive-compulsive disorder[25]. A key emerging theme of this work to date is that whilst patients tend to have more extreme deviations than controls, the spatial distribution of these is extremely heterogeneous, to the extent that group average patterns of neuropathology are simply not representative of most individual patients. However, the exclusive application of normative modelling to structural imaging may have also constrained the potential insight that can be gained. Recent work has shown that these heterogeneous structural deviations are often embedded within shared functional networks[25], emphasising the need to utilise functional imaging measures.

So far no functional neuroimaging-based tool has been meaningfully exploited in clinical practice. One key reason underlying this is that fMRI offers only an indirect measure of neuronal function, remaining abstracted from the cellular and molecular mechanisms that ultimately constitute brain function, and crucially, upon which interventions act. We contend that this may mean that deviations from health characterised using fMRI, as well as the aforementioned structural measures, are not readily amenable to targeted clinical intervention. However, a novel suite of analytic approaches which incorporates micro-scale molecular information into the analysis of macro-scale fMRI dynamics offer critical opportunities to bridge the gap between these scales, providing new insights into brain function and dysfunction (see ref. 33 for extensive review). These approaches are well suited to provide novel biomarkers that link neuropathology through to pharmacotherapy in a mechanistic and data-driven manner[33]. For example, Receptor-Enriched Analysis of functional Connectivity by Targets (REACT) has proven useful in characterising the complex psychopharmacological effects of various drugs[33–36]. Furthermore, it is increasingly being applied to clinical conditions, such as within a recent proof of concept paper demonstrating its potential to stratify patients with osteoarthritis who may respond preferentially to placebo or duloxetine[37]. Crucially, REACT can be used to derive molecular-enriched networks capturing the relationship between receptor density distribution and functional connectivity

(FC) patterns. Specifically, molecular-enriched FC provides an indication of how each brain region, or voxel, interacts with brain areas where a certain receptor is highly distributed, providing a framework for understanding how specific neurotransmitter systems, identified by their receptor distributions, influence brain connectivity patterns. It provides a unique perspective on the functional architecture of the brain, suggesting how neurotransmitter-specific networks could modulate functional connectivity. However, it is important to note that since BOLD fMRI has no intrinsic selectivity to any neurochemical target, it does not directly measure neurotransmitter activity, nor does it imply changes in neurotransmitter levels. Despite this, it can be deployed within a highly scalable manner to large fMRI datasets, offering invaluable insights into the functional alterations underpinned by underlying molecular mechanisms.

Both normative modelling and molecular-enriched analyses offer substantial promise to overcome two of the pre-eminent limitations in biomarker development; namely, complexity in the form of heterogeneity and the multiscale organisation of the brain. As such, their combination offers a potential path forward for both mechanistic elucidation as well as to help bring neuroimaging closer to clinical implementation. To this end, here we utilised REACT to derive networks enriched within the main modulatory (noradrenaline, dopamine, serotonin, and acetylcholine), excitatory (glutamate), and inhibitory (GABA) neurotransmitter systems within two datasets. We subsequently generated normative models of these molecular-enriched networks across the healthy ageing spectrum. We then examined clinical and functional similarity within and across patients suffering from schizophrenia (SCHZ), bipolar disorder (BPD), and attention-deficit hyperactivity disorder (ADHD). Finally, we used a transdiagnostic deviation-symptom mapping approach to link the symptomatology across all patient groups with their brain's functional deviations from normality and highlight the molecular circuits driving such abnormality. Altogether, this offers significant progress towards generating novel biomarkers transcending multiple organisational scales of the brain and conventional diagnostic compartmentalisation, which in the longer term will offer a tantalising opportunity to link targeted treatment through to specific domains of neurobehavioral dysfunction (Fig. 1).

## Results
### Demographics
Following quality control exclusions, imaging data of a total of 607 healthy subjects (CamCAN: $N = 496$, Mean age (SD) = 52.3 (18.3), M/F = 255/241; UCLA: $N = 111$, Mean age (SD) = 31.2 (8.7), M/F = 60/51) were included in the analysis. From the UCLA dataset, a total of 119 patients was used in the final analysis, including patients suffering from SCHZ ($N = 38$, Mean age (SD) = 35.1 (8.8), M/F = 32/6), BPD ($N = 44$, Mean age (SD) = 34.6 (8.9), M/F = 24/22), and ADHD ($N = 37$, Mean age (SD) = 32.2 (10.1), M/F = 19/18).

### Between-subject symptom similarity
To first explore the phenotypic data, we examined each of the 28 different symptom scores within and across the different clinical groups. The distribution of scores for each of the groups overlapped for most symptoms, with only a few such as BPRS positive symptoms showing clear divergence across the conventional diagnostic groups (SI-Fig. 1). To quantify how similar each patient's constellation of symptoms was to every other patient (Fig. 2), we compared within- and between-group similarity for each diagnosis utilising non-parametric repeated measures ANOVAs and post-hoc tests. These revealed significant higher-level results for SCHZ ($\chi^2 = 42.0$, $p < 0.001$), BPD ($\chi^2 = 6.8$, $p = 0.032$), and ADHD ($\chi^2 = 37.2$, $p < 0.001$) (for descriptive statistics, see SI-table 1). Strong within-group similarity was seen both for SCHZ and ADHD groups, with SCHZ patients showing significantly greater within-group similarity than between-group similarity to BPD ($t = 3.9$, $p_{bonf} < 0.001$) and ADHD ($t = 6.4$, $p_{bonf} < 0.001$) groups. Additionally, SCHZ-BPD similarity was significantly different than SCHZ-ADHD ($t = 2.52$, $p_{bonf} = 0.041$), with SCHZ and ADHD patients showing opposite patterns of symptoms (i.e., negative correlation values). ADHD patients showed significantly greater within-group similarity than between-

**a | Data**

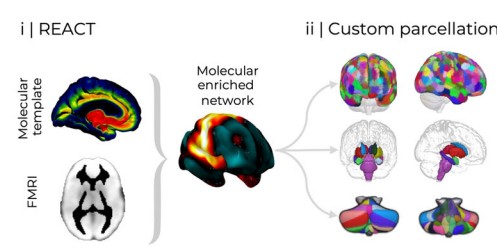

HC
N = 111 UCLA +
496 CamCAN

BPD
N = 44 UCLA

SCHZ
N = 38 UCLA

ADHD
N = 37 UCLA

**b | Psychometric analysis and dimensionality reduction**

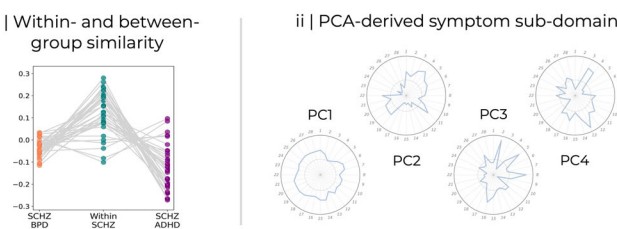

i | Within- and between-group similarity

ii | PCA-derived symptom sub-domains

PC1  PC3
PC2  PC4

**c | Molecular-enriched networks**

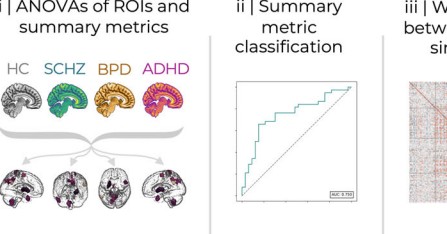

i | REACT

Molecular template

FMRI

Molecular enriched network

ii | Custom parcellation

**d | Normative modelling molecular-enriched networks**

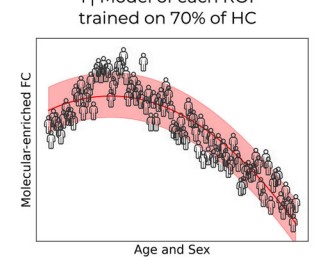 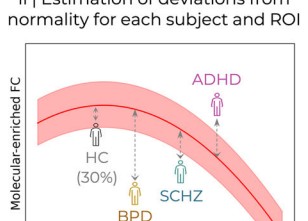

i | Model of each ROI trained on 70% of HC

ii | Estimation of deviations from normality for each subject and ROI

**e | Analyses between diagnostic groups**

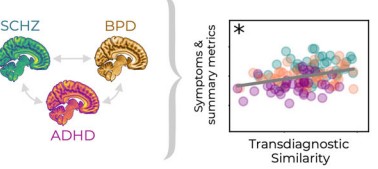

i | ANOVAs of ROIs and summary metrics

ii | Summary metric classification

iii | Within- and between group similarity

HC  SCHZ  BPD  ADHD

**f | Transdiagnostic analyses**

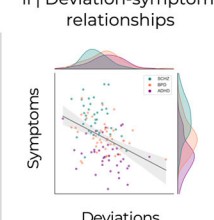

i | Transdiagnostic similarity and correlations with deviation scores and symptoms

ii | Deviation-symptom relationships

SCHZ  BPD  ADHD

**Fig. 1 | Analysis overview. a** The data utilised is from two existing datasets: Cam-CAN ageing and UCLA phenomics, the latter including healthy individuals as well as data from three clinical cohorts (Schizophrenia - SCHZ, bipolar disorder - BPD, and ADHD). **b** 28 different symptom scores available transdiagnostically were used to examine within- and between-group similarity in terms of symptomatology (I) as well as derive composite symptom sub-domains using PCA (ii). **c** REACT was used to generate molecular-enriched functional networks (i) which were subsequently parcellated using a custom combination of cortical, subcortical, and cerebellar ROIs (ii). **d** These ROI-based molecular-enriched networks were then used to create normative models trained on 70% of the HC subjects from both CamCAN and UCLA (i), then used to characterise deviations from normality within the remaining 30% of HC as well as the three clinical cohorts (ii). **e** Deviation scores within each brain region and averaged across brain regions as summary metrics were compared using ANOVAs (i); the averaged summary metrics were also examined for classification value using binary logistic regression (ii); within- and between-group FC deviation similarity was also evaluated (iii). **f** Finally, deviation scores were analysed transdiagnostically, examining how transdiagnostic similarity relates to the extent of deviation and composite symptom sub-domains identified in B (i). Regional deviations were also correlated with symptom sub-domains (ii).

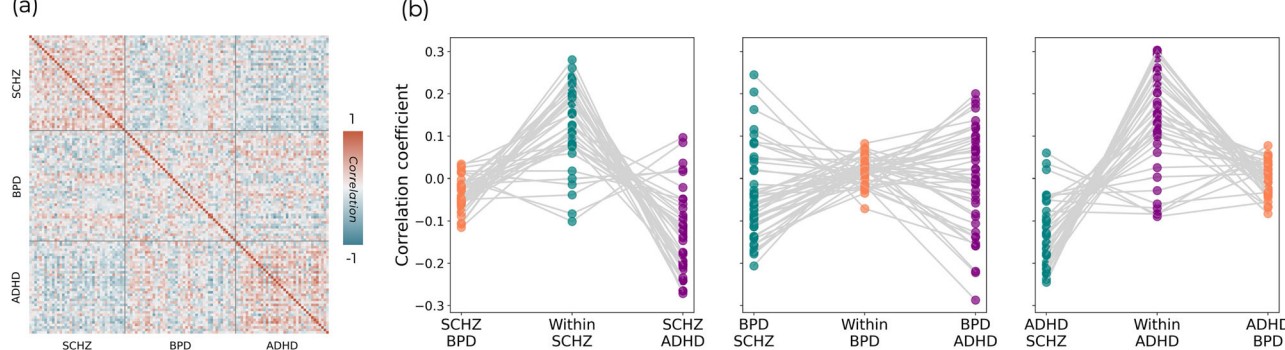

**Fig. 2 | Symptom similarity. a** Between-subject similarity matrix for normalised psychometric measures. Each position in the matrix represents the correlation coefficient across all psychometric scores for a pair of individuals. Grey bars separate the conventional diagnostic criteria, delineating matrix regions of within- (diagonal) and between-group (off-diagonal) similarity. **b** Plots of average similarity of each patient to those with the same diagnosis (within-group) and those with the other two diagnoses (between-group) for each diagnostic group.

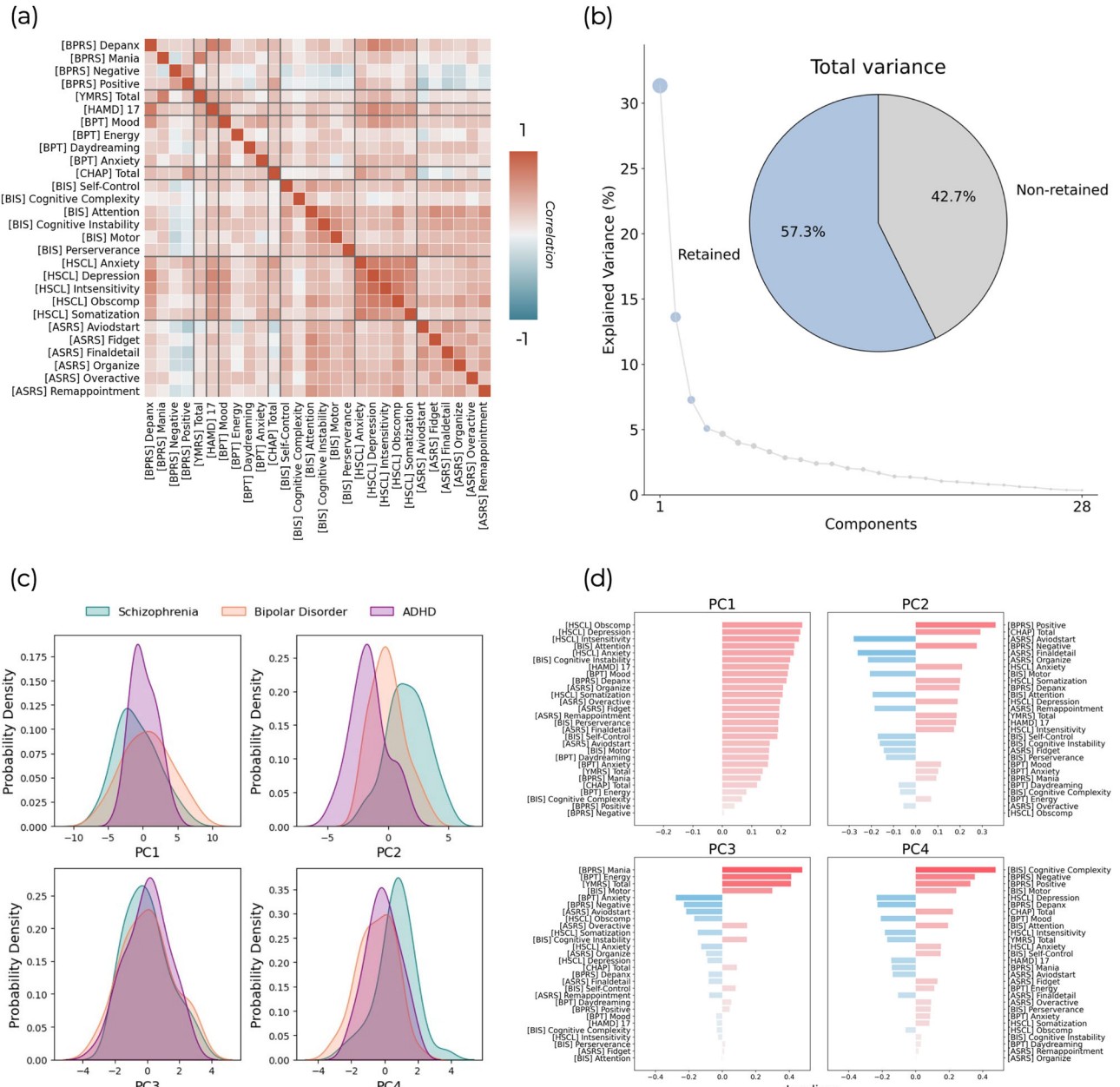

**Fig. 3 | Symptom dimensionality reduction. a** Cross-correlations between each pair of psychometric measures. **b** Component selection. Four components were retained which explained 57.3% of the variance in the original symptom scores. **c** The scores for each component across individuals within the three different clinical cohorts. **d** The loadings of the different symptom scores onto these four components are shown as bar plots.

group similarity to SCHZ ($t = 6.0$, $p_{bonf} < 0.001$), but not BPD ($t = 2.3$, $p_{bonf} = 0.069$). ADHD-SCHZ similarity was significantly lower than the ADHD-BPD one ($t = 3.72$, $p_{bonf} = 0.001$). On the contrary, the post-hoc comparisons for BPD showed that within-group similarity was only marginally greater than between-group similarity with SCHZ ($t = 2.5$, $p_{bonf} = 0.049$) and statistically comparable to between-group similarity with ADHD ($t = 0.5$, $p_{bonf} = 1.0$). BPD-SCHZ and BPD-ADHD similarities were not significantly different ($t = 2.03$, $p_{bonf} = 0.138$). Altogether, SCHZ and ADHD broadly showed stronger within- than between-group similarity, whilst BPD patients showed a pattern of similarity with either SCHZ or ADHD patients.

**Transdiagnostic psychometric dimensionality reduction**
Across the 28 available psychometric scales and sub-scales there was generally strong overlap between clinical conditions. Cross correlations

between these normalised psychometric measures showed a mixture of positive and negative relationships (Fig. 3a), with the strongest values representing those which largely measure the same construct (e.g., BPRS depression and anxiety subscale correlates strongly with the total Hamilton depression score) and negative relationships between those measuring conceptually diverging constructs (e.g., the Adult Self-Report Scale primarily assesses ADHD symptomatology and shows mostly negative correlations with the Chapman Scales of psychosis proneness). Given the number of highly correlated measures, we utilised principal components analysis (PCA) on these psychometric variables across all patients, resulting in four variables which met eigenvalue one criterion and explained 57.3% of the total variance (Fig. 3b). These component scores retained significant overlap across the diagnostic groups (Fig. 3c), although PC2 showed a clearer separation across them than most of the original psychometric distributions (Fig. 3a). The first principal component (PC1) represented

general psychopathology with positive loadings across all psychometric measures (Fig. 3d). The subsequent 3 components showed more specific patterns of positive and negative loadings. PC2 captured a unified dysregulation and impulsivity framework encompassing psychotic and depressive symptomatology, with strongest loadings for positive and negative symptoms, psychosis proneness, as well as depression and anxiety, but also some inattention sub-scores (i.e., avoid start, detail, and organisation) of the Adult Self-Report Scale and the motor impulsiveness items of the Barratt Impulsiveness Scale as negative loadings. PC3 captured a constellation of manic symptoms, with very strong loadings for mania (both the BPRS and YMRS), energy, and motor impulsiveness. Interestingly, there were also negative loadings for the Adult Self-Report Scale of ADHD symptoms, with the exception of overactivity. Finally, PC4 captured cognitive complexity and psychotic symptoms (both positive and negative symptoms), but unlike PC2, the loadings for depressive symptoms were negative. Given the strongest loading is for cognitive complexity, this component may be separating patients that show relative preservation from cognitive decline. Additionally, this component shows positive loadings across the impulsivity measures which are all negative in PC2.

## Molecular-enriched networks

We utilised REACT, as summarised in SI-Fig. 2, to produce a set of molecular-enriched networks for the CamCAN and UCLA healthy subjects as well as UCLA clinical cohorts. In essence, these networks capture the spatiotemporal relationships between fluctuations in the BOLD signal and the spatial distribution of different molecular systems delineated by PET/SPECT templates (subsequently referred to as molecular templates) derived from independent cohorts of healthy controls. This is achieved using a two-step multiple regression analysis which yields one map of molecular-enriched FC for each subject and molecular system that can be utilised for subsequent analyses. Within these molecular-enriched FC maps, positive values reflect stronger coupling to the dominant BOLD fluctuations within the distribution of a given molecular system whilst negative values represent anti-correlation to the dominant BOLD fluctuations within the distribution of a given molecular system. The details of REACT are discussed at length within[33].

These networks are shown averaged across healthy individuals from both datasets in SI-Fig. 2b. The correlation coefficients between the different molecular templates used in the REACT analysis (SI-Fig. 2c) as well as between the molecular-enriched networks (SI-Fig. 2e) were moderate, with generally stronger overlap within the neuromodulatory and excitatory/ inhibitory neurotransmitter groups than between them. VIF values between the different molecular templates were also moderate (SI-Fig. 2d), but importantly not surpassing the rule-of-thumb value of 5 above which issues of collinearity are considered sufficiently high to warrant additional consideration. Finally, the cross-correlations between the resulting molecular-enriched (SI-Fig. 2e) networks showed a similar pattern to the cross-correlations of the molecular templates from which they were derived (SI-Fig. 2C), again showing generally stronger overlap within the neuromodulatory and excitatory/inhibitory neurotransmitter groups than between them. Overall, these networks capture different parts of the BOLD signal relating to the spatial distribution of different molecular systems.

## Modelling the normative molecular-enriched brain

Normative models were trained on 70% of the healthy control data across both UCLA and CamCAN datasets and tested on the remaining 30%. These trained models explained a moderate amount of variance (EV) in molecular-enriched networks in this unseen data using age and sex as predictor variables. This EV varied substantially between molecular enriched-networks and across ROIs (SI-Fig. 4). Herein, all analyses utilise only regions that had positive EV for that receptor system (from the total of 443 regions, this was 319, 307, 296, 257, 418, and 421 for NAT, DAT, SERT, VAChT, mGluR5, and GABA-A respectively). This reflects the spatial heterogeneity of the molecular-enriched networks, with each system being associated with key nodes in the brain and poorly associated with other regions for which the normative models failed to converge on a stable estimate.

## Comparing deviations from normality across conventional diagnostic groups

Our normative modelling produced deviation scores for the healthy subjects and patients (one deviation value per ROI, subject, and molecular system, which are shown averaged in SI-Fig. 3). These deviations, represented as either positive or negative values, indicate how much a subject's molecular-enriched FC for a specific brain region deviates from the established normative FC model derived from healthy controls. A positive deviation value in a certain ROI indicates that its' functional coupling with areas of high receptor density is more pronounced than typically observed, reflecting potentially more synchronous or integrated activity of this ROI within the molecular-enriched network. Conversely, negative values indicate that the ROI's FC with areas of high receptor density is reduced, suggesting a decrease in synchronisation or integration within these receptor-rich regions compared to the norm. It is crucial to note that while these deviations in molecular-enriched FC reveal variations in connectivity patterns, they do not directly measure or imply changes in neurotransmitter activity. Instead, these metrics help us understand how networks, hypothesised to be modulated by specific neurotransmitter systems based on receptor density, function in comparison to a normative framework. These deviation scores were compared across groups for each molecular system with $1 \times 4$ ANOVAs. The tests revealed statistically significant differences within the VAChT and mGluR5 systems (Fig. 4a). Lower-level t-tests revealed that the VAChT result was driven by differences between $HC_{UCLA}$ and SCHZ, with a mixture of regions where patients had molecular-enriched FC values greater or lower than the normal population ranges, including the left putamen, left angular gyrus, bilateral precuneus, left supplementary motor area (SMA), right hemispheric lobule IX, and vermal lobule VIIIb. Post-hoc tests on the deviations related to the mGluR5-enriched network showed differences between $HC_{UCLA}$ and both SCHZ and BPD (Fig. 4b), with differences largely being in the direction of FC values lower than the normal range for these two clinical cohorts. Significant clusters were localised in vermal lobule VI and the right mid-occipital cortex (Fig. 4b). It is also worth noting that the mGluR5 system had many clusters just below the significant threshold, including the right insula ($F = 6.13$, $p = 0.059$ and SMA ($F = 5.86$, $p = 0.059$). No significant differences were found between $HC_{UCLA}$ and the ADHD group in any of the molecular-enriched networks. When considering differences between the clinical groups, lower-level comparisons additionally revealed that within the cholinergic system, BPD patients had greater deviations than ADHD patients in the left precuneus and that SCHZ patients had greater deviations than ADHD patients within bilateral precuneus, left SMA, left angular gyrus, left pallidum, right hemispheric lobule IX, and vermal lobule VIIIb, similarly to what seen between SCHZ and $HC_{UCLA}$. For the glutamatergic system, SCHZ patients showed significantly greater deviations from the normal range than ADHD within the vermal lobule VI. Overall, these results highlighted between-group differences largely driven by SCHZ for the VAChT-enriched network and both SCHZ and BPD for the mGluR5-enriched network, and indicated that the range of deviations within ADHD is comparable to those seen in the healthy controls.

To examine whether we could produce a summary metric characterising each subject's deviations within a given system rather than having one value for every ROI, we computed the mean deviation values across regions and statistically compared these across the different groups of participants using $1 \times 4$ ANOVAs. We found significant differences across groups for the summary deviation metric related to mGluR5- ($F = 4.62$, $p < 0.001$) and GABA-A-enriched networks ($F = 3.24$, $p = 0.02$; SI-Fig. 5). Lower-level Tukey honestly significant difference tests revealed significant differences in summary deviation metrics for mGluR5-enriched networks only between SCHZ and healthy controls ($p_{corr} = 0.003$) as well as BPD and healthy controls ($p_{corr} = 0.022$). Similarly, lower-level comparisons for GABA-A-enriched network's summary metrics showed significant differences for SCHZ compared to healthy controls ($p_{corr} = 0.014$). All these differences indicated a lower molecular-enriched FC in patients than healthy controls, in line with the broadly negative t values across the brain seen in the ROI-

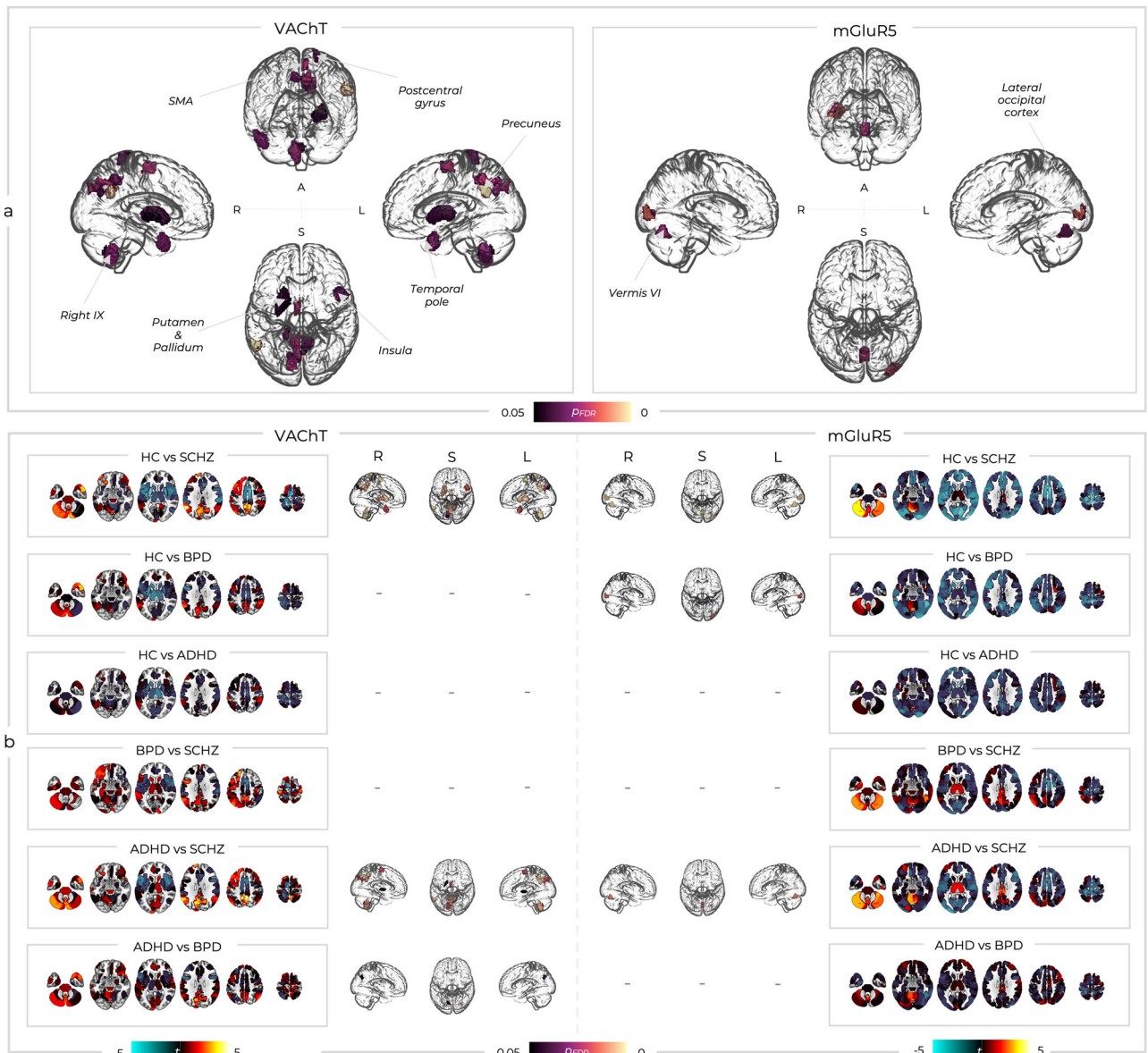

**Fig. 4 | Between-group differences in deviation scores. a** Only two molecular-enriched networks had significant ($p_{FDR}$) between-group differences (A; anterior view, L; left view, R; right view, S; superior view). SMA; supplementary motor area, STG; superior temporal gyrus. **b** Lower-level t-tests showed which between-group comparisons were driving the higher-level results. t statistic colour bars correspond to the order of the sub-box titles such that when the left group has greater deviations these are blue and when the right group has greater deviations these are red/yellow (A; anterior view, L; left view, R; right view, S; superior view).

wise analysis (see Fig. 4b). The summary deviation metrics that showed significant differences in lower-level comparisons were then used within a binary logistic regression, which revealed a moderate capacity to discriminate patients with SCHZ (SI-Fig. 5a/c) and BPD (SI-Fig. 5b) from healthy controls. Specifically, the mean mGluR5-enriched network summary deviation metrics had an area under the curve (AUC) of 0.75 for SCHZ and 0.66 for BPD. Similarly, the mean GABA-A-enriched network summary deviation metrics resulted in an AUC of 0.73. Overall, this analysis revealed that summary metrics of some of the networks under exam, including those related to GABA-A and mGluR5, are useful markers of deviation from normality for SCHZ and BPD.

### Between-subject FC deviation similarity
As done for the symptom scores, we examined the between-subject correlations across all ROIs deviation scores for each molecular-enriched functional network (Fig. 5a) and calculated the within-group similarity of FC deviations from normality by averaging each individual's similarity with the

other individuals of the same group. Between-subject correlations were calculated within each molecular system (Fig. 5a), providing a measure of how similar subjects' deviation scores are across individuals in the same group and between different groups. The healthy controls were consistently dissimilar from each other, with the distribution of their within-group similarity centred around zero (Fig. 5b). Kolmogorov-Smirnov tests revealed that the distribution of within-group similarity in SCHZ and BPD were significantly different to those in $HC_{UCLA}$ for all molecular-enriched networks (SI-table 2), generally showing greater mean within-group similarity in patients than $HC_{UCLA}$ (SCHZ > BPD > ADHD > $HC_{UCLA}$). The ADHD group showed significant differences from the $HC_{UCLA}$ distribution for NAT and DAT, but not for the other molecular-enriched networks (SI-table 2), although none of these ADHD results survived Bonferroni correction ($p < 0.05$ / 18).

We also estimated the between-group similarity for each clinical cohort, i.e., the mean FC deviation similarity of each patient with individuals of different clinical cohorts, and compared within- and between-group

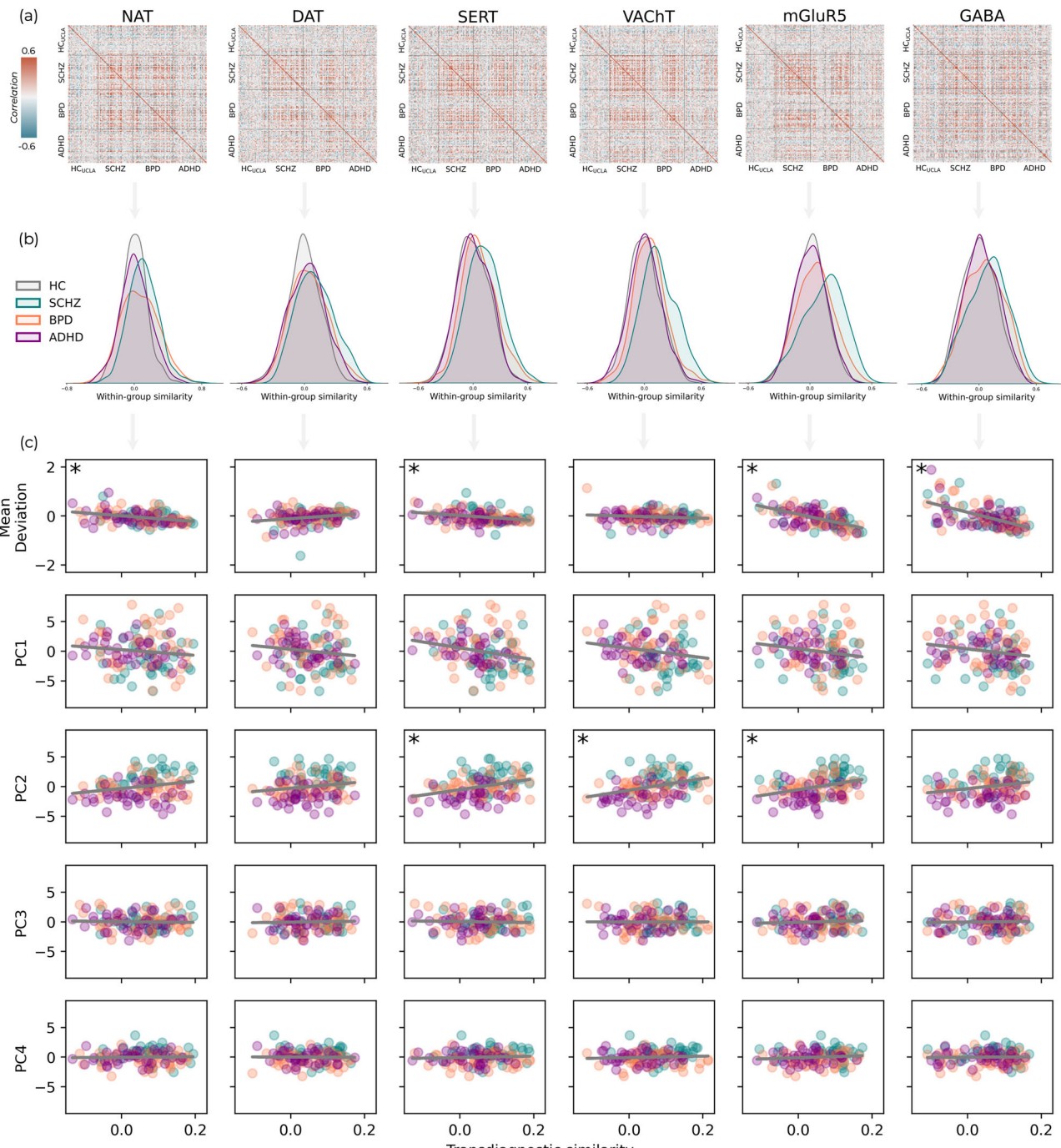

**Fig. 5 | Deviation similarity within and between groups. a** Matrices of between-subject correlations of deviation scores across ROIs for each pair of individuals within and between groups. **b** The correlation coefficients for within-group similarity are displayed as density plots. **c** The relationship between patients' similarity to the other patients across all diagnostic groups and the overall deviation burden categorised by mean deviation score across their whole brain (top row) as well as each symptom component. Asterisks denote relationships that are significant following Bonferroni correction ($p < 0.05/30$).

similarity scores using non-parametric repeated measures ANOVAs and post-hoc comparisons (SI-table 4). SCHZ consistently showed greater within- than between-group similarity both with BPD and ADHD groups, whilst both BPD and ADHD showed comparable within- and between-group similarities, with few exceptions (within-group similarity of BPD was significantly different than BPD-ADHD similarity for SERT- and VAChT-enriched networks; within-group similarity of ADHD was significantly different than between-group similarity with BPD and SCHZ for the SERT-enriched network, and with SCHZ for the mGluR5-enriched network). Overall, unlike the psychometry, ADHD patients showed lower within-

than between-group similarity, indicating that brain's FC deviations do not present any homogeneous patterns within this clinical cohort, but present some similarities with the other clinical groups.

**Transdiagnostic similarity of deviation scores**

After calculating transdiagnostic similarity of FC deviation scores by averaging each patient's similarity to the other patients, disregarding diagnostic labels, we explored the link between these patterns and the summary deviation metrics presented in 2.6 to measure the link between transdiagnostic similarity and degree of FC deviation from normality. Significant

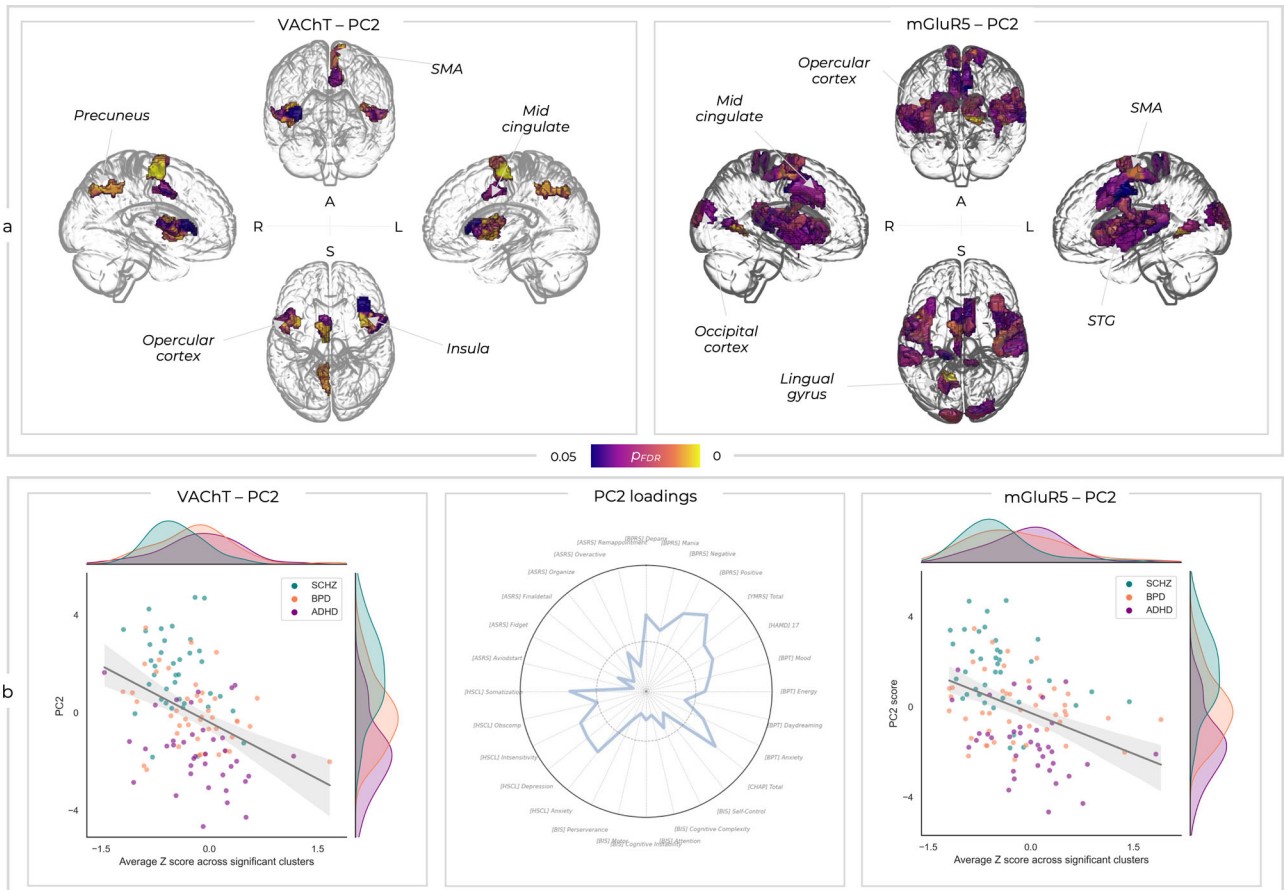

**Fig. 6 | Mass univariate relationships between deviation and symptom scores.** **a** Only two of these deviation-symptom relationships were significant. Significant $p_{FDR}$ values are shown for the relationships between PC2 and deviations within the VAChT- and mGluR5-enriched networks (A; anterior view, L; left view, R; right view, S; superior view). SMA; supplementary motor area, STG; superior temporal gyrus. **b** The same relationships are shown in the brain plots, with Z values averaged across the significant clusters and correlated with PC2 symptom scores. The loadings for PC2 are shown again here for context. Of note, whilst the diagnostic groups are reported in different colours within the scatter plots, this is purely for visualisation purposes and these analyses were run transdiagnostically across all patient groups.

negative correlations were observed between summary deviation metrics and transdiagnostic similarity across networks involving NAT, SERT, mGluR5, and GABA-A, even after applying Bonferroni corrections (Fig. 5c). In other words, individuals with greater transdiagnostic similarity tend to have abnormally lower FC values compared to those typically seen in healthy individuals. This result indicates a link between FC deviation patterns shared across neuropsychiatric patients and the severity of their neuropathological conditions, irrespective of their diagnostic labels.

We further investigated the association between the degree of transdiagnostic similarity of deviation scores and symptom sub-domains, hypothesising that patterns of FC deviations shared across disorders could be linked to similar symptomatology. Significant correlations following Bonferroni correction were found between PC2, i.e., the component capturing psychotic, depressive, inattentive and, impulsive symptomatology, and SERT, VAChT, and mGluR5, highlighting greater transdiagnostic similarity being related to greater PC2 scores (Fig. 5c and SI-table 3). No significant correlations were found between transdiagnostic similarity and the other PCs.

**Transdiagnostic deviation-symptom mapping**
We examined the relationships between the four PCA-based transdiagnostic symptom sub-domains and FC deviations from normality within each ROI and molecular-enriched network. Varying patterns of both positive and negative deviation-symptom relationships were seen across enriched-networks and components (SI-Fig. 6). However, only the relationships between the component associated with the complex

dysregulation symptomatology (i.e., PC2) and the FC deviations from normality in the VAChT- and mGluR5-enriched networks were significant (Fig. 6b). VAChT-enriched FC deviations showed significant relationships with PC2 spanning bilateral insular and opercular regions as well as the left mid-cingulate, SMA, and precuneus. The mGluR5-enriched FC showed relationships with PC2 spanning similar bilateral insular and opercular regions, as well as the bilateral superior temporal gyri (STG), bilateral cuneus, left lingual gyrus, bilateral mid-cingulate, and bilateral SMA. Both these relationships were negative, such that higher symptom scores were associated with molecular-enriched FC lower than the normative population (Fig. 6c). Overall, these findings indicate that irrespective of diagnostic classification, there is a link between variations from typical FC levels in the VAChT- and mGluR5-enriched networks and transdiagnostic sub-domain encompassing psychotic, depressive, inattention, and motor impulsiveness symptoms.

**Discussion**
Here, we bring together two novel classes of neuroimaging analytics which hold significant potential to circumvent key barriers to the neurobiological characterisation and mechanistically informed treatment of neuropsychiatric disorders. Our normative models performed comparably to previous work on structural neuroimaging data, but additionally captured both functional and molecular facets of the brain. Normative modelling revealed more robust between-group similarities and deviation-symptom relationships than conventional REACT analyses, emphasising the power of describing neuropathology as divergence from estimates of healthy ageing.

Our between-group findings converge to broadly implicate the glutamate and GABA in SCHZ and BPD (possibly reflecting altered excitatory-inhibitory balance), the cholinergic system in SCHZ, and largely failed to find specific disorder-based alterations in ADHD. However, moving beyond these categorisations, we found substantial similarity between individuals transdiagnostically and that deviations mapped onto symptom scores across groups. Interestingly, the transdiagnostic sub-domain we identified with PCA, which encompasses psychotic, depressive, inattentive, and motor impulsive symptoms across SCHZ, BPD, and ADHD, was able to capture a new, broad symptomatology spectrum within which patients with different diagnostic categorisations are situated. Furthermore, these transdiagnostic symptoms were linked to abnormal variations from typical FC levels in the VAChT- and mGluR5-enriched networks. Herein, we discuss these key findings in light of the challenges presented by the multiscale organisation of the brain and heterogeneity within clinical neuroscience.

Despite departing from the typical use of structural imaging features, our models explained a comparable amount of variance to a recently published paper which modelled grey matter volume[24], despite the fact that functional imaging data is higher dimensional and noisier, which translates into a greater variability. This constitutes a significant step forward for normative modelling, suggesting that this nascent field can begin to additionally exploit the rich spatiotemporal dynamics of functional imaging, which may offer benefits over and above measures of grey and white matter in characterising neurobiological heterogeneity. Our results move beyond both conventional structural and functional MRI measures, which are inherently incapable of providing insights into their cellular and molecular underpinnings. The use of molecular-enriched functional imaging approaches offers a tantalising opportunity to circumvent these barriers and make use of already abundant fMRI data to provide insights spanning the molecular and systems levels[33]. Crucially, this enables a non-invasive characterisation of dysfunction that is directly amenable to targeted pharmacotherapeutic intervention in a scalable manner. This could be achieved through a broad categorisation of which molecular-enriched networks are altered by acute drug challenges with different compounds, allowing these to be targeted in patients showing substantial deviations within networks enriched with the same molecular targets. This work offers a provisional proof-of-concept set of results that should be further expanded with more comprehensive datasets, e.g., including treatment responsiveness required to meaningfully test these findings.

Our between-group analyses broadly converged upon the glutamatergic and GABAergic systems, potentially reflecting excitatory-inhibitory (E/I) imbalance, which is a widely implicated facet of neuropathology in SCHZ and BPD[38], and provides confidence in our methods. Across all our analyses comparing deviations between SCHZ and controls we identified differences in the glutamatergic and/or GABAergic systems, with patients broadly showing FC values lower than the normative ranges and greater similarity to one another than healthy individuals. Within SCHZ, various aspects of glutamatergic and GABAergic (dys)function are evidenced by key risk loci within genetic studies[39–42], altered resting gamma power[43–46], sensory gating deficits[47–49], changes in various TMS-EEG paradigms[50], reduced mismatch negativity amplitude[51,52], reduced hippocampal of NMDA and GABA-A receptor density[53,54], lower post-mortem levels of the GABA synthesising enzyme glutamate decarboxylase 67[55–57], and parvalbumin-positive interneurons in SCHZ[58–62]. Although some of these measures have also shown null findings in additional studies, this critical mass of literature strongly implicates E/I imbalance within SCHZ[2,38,63,64]. We further add to this overall picture with results relating to both glutamate and GABA, supporting the idea that E/I imbalance in SCHZ emerges from an interplay of these two systems, as opposed to one or the other, which remains an outstanding issue within the field[63]. Similarly, the glutamatergic system was consistently implicated in BPD across our analyses. Despite a heavy focus on monoamines for over half a century, the glutamatergic system is increasingly thought to play a key role in mood disorders[65]. A meta-analysis of magnetic resonance spectroscopy studies found that glutamate levels were elevated in BPD compared to controls when all brain areas were combined, regardless of medication status[66]. A follow-up meta-analysis largely corroborated this, identifying increased frontal glutamate and decreased mismatch-negativity[67]. Additional evidence from genetics[68–76], blood and urine markers[77–80], as well as post-mortem studies[81–86] point towards glutamatergic dysfunction. We found no significant differences between glutamatergic or GABAergic deviations within SCHZ and BPD, whilst SCHZ and ADHD showed differences for both mGlur5 and VAChT which mirrored differences between SCHZ and controls. This suggests a general neurobiological similarity between SCHZ and BPD whilst ADHD participants more closely resembled controls. Our results are therefore broadly convergent with previous accounts of E/I imbalance in these disorders as well as some level of shared underlying neuropathology, offering confidence that our normative modelling approach aiming to link molecular- and systems-level mechanisms is capturing meaningful aspects of neuropathology. Furthermore, the failure of our conventional REACT analyses to identify any between-group differences emphasises the value of describing pathology as divergence from normative values. Moreover, the fact that these differences were captured by whole-brain mean deviation summary scores provides tentative support that this approach could be used to create simplified clinical readouts of the integrity excitatory and inhibitory molecular-enriched networks.

The cholinergic findings relating to SCHZ throughout our results are intriguing. All currently approved antipsychotics target the dopamine D2 receptor[87], with actions on ventral tegmental area (VTA) dopaminergic neurons thought to underlie therapeutic efficacy in reducing positive symptoms whilst unwanted action on the substantia nigra results in extrapyramidal side effects[64,88–90]. However, current treatments provide little benefit for negative or cognitive symptoms and many patients continue to experience residual positive symptoms or remain treatment-resistant[91,92]. There is increasing interest in novel cholinergic compounds being able to treat SCHZ symptoms, including positive, negative, and cognitive symptoms, whilst not being associated with the long-term side effects of dopaminergic antipsychotics[93,94]. Anticholinergic drugs can induce or exacerbate confusion, delirium, cognitive impairment, and hallucinations, with this mimicry of psychiatric symptoms indicative of the role cholinergic mechanisms may play in these symptoms when arising clinically[93,95]. Conversely, in the 1990's xanolemine, a muscarinic antagonist being developed for cognitive symptoms of Alzheimer's disease, was noted to have antipsychotic effects[96], and these results have been replicated in SCHZ patients[97–99]. Interestingly, these studies showed benefits for positive and negative symptoms, but also cognitive performance. Crucially, xanolemine's efficacy is not mediated by direct action on D2 receptors[100]. Therefore, pharmacological manipulation of the cholinergic system has been robustly demonstrated to induce or ameliorate the symptoms of SCHZ. Our findings add credence to this view, suggesting that the cholinergic system is related to altered systems-level dynamics in SCHZ compared to controls and that individuals with more negative cholinergic deviations had greater PC2 scores, which reflect core SCHZ symptoms. It is therefore tempting to speculate that xanolamine may act to normalise the cholinergic dysfunction spanning molecular and systems levels identified here. This may be mediated through indirect actions of the laterodorsal tegmentum on the dopaminergic system[101,102], potentially preferentially acting on VTA dopaminergic neuron firing whilst sparing the substantia nigra, mitigating extrapyramidal side effects and tardive dyskinesia[103]. However, additional preclinical evidence supports the idea that xanolemine may act by modulating glutamatergic microcircuits, which may in turn also impact dopaminergic transmission[104–106]. These complex receptor sub-type-specific mechanisms have been discussed at length recently[93,94,107]. However, a key point is the emerging idea that novel cholinergic compounds could provide antipsychotic effects through conventional dopaminergic pathways thought to drive positive symptoms, but additionally benefit negative and cognitive symptoms not ameliorated by current treatments[93,94]. Our PC2 relates to a broad range of psychotic symptoms, potentially supporting links through to the cholinergic system regardless of diagnosis.

The lack of results for ADHD throughout our analyses is generally consistent with this disorder being extremely heterogeneous. Indeed, previous work utilising normative models of grey and white matter volume found that almost no brain regions showed consistent deviations within ADHD patients[30] and a recent meta-analysis of 96 structural and functional imaging studies in ADHD found a lack of regional convergence[108]. Similarly, we identified no significant differences between ADHD subjects and healthy controls within any brain region nor in our summary measures of mean deviation scores. Intriguingly, we did find significant differences in the distribution of between-subject similarities for ADHD participants and controls for the noradrenergic and dopaminergic-enriched networks, although distributional differences were small and did not survive Bonferroni correction. The catecholamines are long implicated in the pathophysiology and treatment of ADHD[109,110], adding credence to these similarity analyses which may offer an alternative lens through which to view the pattern of deviations across subjects. Analogous to the recent combination of normative modelling and network-lesion mapping, wherein deviations are examined for co-localisation to common networks[25,111], it would be interesting to examine whether deviations within ADHD map onto regions strongly influenced by catecholaminergic transmission. This could be indexed through connectivity patterns of the noradrenergic locus coeruleus and dopaminergic ventral tegmental area, the distribution of different catecholaminergic receptors and transporters, or some multi-modal combination of these converging measures. A larger sample focussed explicitly on characterising ADHD deviations and symptomatology would be better placed to investigate this possibility. However, we cannot exclude the possibility that ADHD is associated with more subtle differences in neural measures such as molecular-enriched functional networks, which are harder to detect than potentially more substantial pathophysiology within SCHZ and BPD.

Despite receiving different diagnoses, we found transdiagnostic similarities across groups within our analyses. Patients' symptom scores were broadly correlated regardless of diagnosis, indicative of highly overlapping clinical phenotypes and this was substantially recapitulated in the deviation scores. Overall, SCHZ patients broadly showed greater within- than between-group similarity, whilst BPD and ADHD patients showed a more mixed profile, emphasising the highly overlapping nature of both symptoms and neurobiology, reflecting the fact that current diagnostic labels to not provide sufficient precision to fully differentiate groups or robustly target treatment.

The level of transdiagnostic deviation similarity for noradrenergic, serotonergic, glutamatergic, and GABAergic systems was inversely related to the summary deviation scores, indicating that those who were more similar had greater negative deviation, i.e., FC values lower than the normal ranges of the healthy population. Likewise, the level of transdiagnostic similarity in terms of FC deviations in the serotonergic, cholinergic, and glutamatergic functional networks was related to strong symptom scores related to psychotic, depressive, inattention and impulsiveness symptom domains (PC2). These within-group and transdiagnostic similarity analyses may also relate to a general body of literature which suggests that neuropathology associated with various disorders of the brain may restrict the repertoire of network states a brain can inhabit[112–114]. When considering direct relationships between deviations and our dimensionally reduced symptomatology components, we found transdiagnostic relationships further linking these PC2 symptom domains through to cholinergic and glutamatergic network deviations. Importantly, this symptom spectrum captures positive loadings from positive and negative psychotic symptoms and depressive symptoms, as well as negative loadings from the adult self-report scale of ADHD symptoms, creating a symptom spectrum upon which individual patients can be situated according to their specific symptoms, regardless of diagnosis. These similarity-symptom and deviation-symptom mapping approaches offer innovative approaches to delineate novel transdiagnostic symptom-network spaces which may form the basis for novel biomarkers in the longer term.

It remains unclear why we found cholinergic and not dopaminergic results across these analyses, given the core role dopamine is thought to play within the pathophysiology of SCHZ[64,88–90]. The distribution of dopaminergic and cholinergic transporter density strongly overlaps within key regions such as the striatum. It is therefore possible that our cholinergic results additionally reflect dopaminergic mechanisms, especially given how strongly mechanistically intertwined these systems are in shaping cortico-striato-thalamic circuitry[115]. Moreover, there is high VAChT density within the thalamus, another region implicated in the pathophysiology of SCHZ[116,117], potentially contributing to the cholinergic findings. Regardless of the precise delineation between the two systems, these regions which differ between groups and relate to transdiagnostic symptomatology are influenced by the cholinergic and dopaminergic systems, with neurobiologically plausible accounts for cholinergic dysfunction giving rise to multiple symptoms in SCHZ. Future methodological development will be crucial to carefully tease apart relationships where receptor density and pathophysiology are highly colocalised and link these measures through to treatment. Future work could consider examining deviations within dopaminergic, cholinergic, glutamatergic, and GABAergic receptor subtypes to allow for clearer delineation between putative drug targets and core symptom domains. Alternatively, focusing on results which converge with other methodologies such as using seed-based connectivity from dopaminergic and cholinergic nuclei may be beneficial. Additionally, the availability of treatment response outcome measures would allow for the direct examination of whether dysfunction in a given molecular system is indeed predictive of treatment responsiveness to an intervention which targets that system. Methodological progress on this front will be crucial to move beyond simplified accounts of neurobiology and neuropathology, considering the full complexity of multiple neurotransmitter systems acting in concert.

This work is not without limitations. As with all molecular-enriched network analyses, we utilise group average molecular templates acquired in separate healthy subjects. The validity of this approach has been described extensively within the broad literature utilising PET/SPECT and transcriptomic data within neuroimaging, as discussed at length in ref. 33. However, it is important to acknowledge these templates were acquired in separate subjects and using different methodologies, resulting in differences in resolution despite being normalised into the same space. Several of these molecular enriched-networks also show moderate collinearity, which requires careful consideration in the multiple-regression approach employed within REACT. However, we show that VIF values are below 5, suggesting that levels are not broadly problematic in the models employed here. This may present a barrier to the inclusion of additional receptor systems in the future, and finding new ways to examine the full repertoire of molecular systems and sub-systems within the human brain will be important in the longer term. The cross-sectional nature of our data also confines us to observe associations rather than establishing causal relationships between deviations and clinical measures. Longitudinal studies are imperative to delineate the true temporal trajectory of molecular-enriched networks and their relationships to symptomatology as well as characterise their potential diagnostic, prognostic, and treatment-predictive capabilities. Similarly, the sample sizes of the clinical cohorts utilised here were relatively small, with our proof-of-concept results requiring follow-up in more robustly powered samples, ideally also including a diverse set of patients which often fall outside the scope of the strict selection criteria of smaller studies. Additionally, we only tested our deviation-symptom mapping in one dataset. Future work will require a simplified set of transdiagnostic symptom measures such that the same relationships can be tested in replication samples. Finally, we cannot exclude the possibility that treatment within the clinical cohorts may impact the estimation of deviations within the molecular-enriched functional networks, potentially reducing the extent of deviations in those subjects responding well to treatment while inducing deviations in regions of high target engagement but minimal contribution to pathology. Future studies examining drug-naïve populations, or with

sample sizes sufficiently large to attempt to control for treatment type, will be important moving forward. Demonstration that deviations characterised by combining molecular-enriched network analyses and normative modelling are actually predictive of treatment outcomes will be the ultimate test of the extent of these limitations.

The integration of novel functional-molecular neuroimaging techniques, normative modelling, and a transdiagnostic perspective utilised here offers methodological and theoretical progress towards an understanding of the shared neurobiological foundations that underpin psychiatric conditions. Our transdiagnostic approach moves away from case-control analyses and offers an interesting way to situate clinical groups or individuals within between-subject similarity and deviation-symptom landscapes, which when scaled up across diagnoses, symptomatology, and molecular systems may offer novel perspectives on how complex aberrations of affect and cognition map onto dysfunction spanning molecular and systems level readouts. The long-term goal of this approach would be to build analytic bridges between these neuroimaging-derived brain phenotypes and treatments. Regardless of the progress made in diagnostics and prognostics, these both serve the ultimate goal of facilitating the provision of the right treatment to the right patient at the right time. The opportunity to link symptoms to non-invasive measures of molecular mechanisms amenable to pharmacotherapeutic intervention may prove a useful tool for precision psychiatry in the longer term.

## Methods
### Datasets
This study utilises two separate existing datasets. Firstly, the healthy ageing CamCAN dataset (obtained from the CamCAN repository available at http://www.mrc-cbu.cam.ac.uk/datasets/CamCAN/[118,119]) was chosen as it spans the full spectrum of healthy ageing. This allows normative models to provide relatively robust estimates across different test datasets, permitting some level of generalisability for subsequent investigations. Secondly, the UCLA phenomics dataset (ref. 120, available from https://openneuro.org/datasets/ds000030/versions/1.0.0), was selected as it included healthy individuals as well as multiple psychiatric cohorts with deep clinical phenotyping, allowing for examination of deviations from healthy individuals both within and across conventional diagnostic boundaries. Full details of the participants from each dataset as well as the inclusion and exclusion criteria are included in the supplementary materials.

### Clinical and behavioural data
A comprehensive list of the behavioural assessments can be found in Table 3 of the original manuscript[120]. Here, we utilised symptom measures from the Young Mania Rating Scale-C (YMRS), Hamilton Psychiatric Rating Scale for Depression (HAMD-17), Brief Psychiatric Rating Scale (BPRS), Hopkins Symptom Checklist (HSCL), and Adult Self-Report Scale v1.1 Screener (ASRS). Additional trait measures included in our analyses were the Barratt Impulsiveness Scale (BIS), Scale for Traits that Increase Risk for Bipolar II Disorder (BPT), and the Chapman Scale for Perceptual Aberrations (CHAP). Symptom and trait scores selected had data available across all three clinical groups, providing measures of symptoms that are conventionally associated more with one of the diagnostic groups, but with potential involvement within each. Participants with incomplete demographic or psychometric data were excluded. Where relevant sub-scores were available for these symptom and trait measures, we utilised these within subsequent analyses to preserve the rich dimensionality of this phenotypic data. Where sub-scores were not available, we used the total summary score. In total, this offered 28 measures. Each measure was plotted as density curves split by clinical diagnosis to see how overlapping or non-overlapping they were.

We additionally examined within- and between-group similarity of these scores. We created a correlation matrix to examine how correlated each subject was to every other subject across all 28 clinical scores available. Then, we utilised non-parametric repeated measures ANOVAs implemented with Friedman test and Conover's post-hoc comparisons to examine whether each group differed in its within- and between-group similarity, applying Bonferroni correction for multiple comparisons.

### Dimensionality reduction of psychometry
We examined the level of collinearity between the large number of different highly colinear clinical measures available within the UCLA dataset associated with deep phenotypic characterisation. Given the high levels of collinearity observed, we utilised principal components analysis (PCA, implemented in Python using sklearn) to reduce the 28 different psychometric sub-scores and summary scores into a smaller set of components explaining a substantial portion of the variance in these scores across subjects. The psychometric data was normalised (using sklearn.preprocessing.StandardScaler) prior to being entered into the PCA and no rotation was employed. Importantly, we pooled together the scores across the three clinical cohorts to examine this symptomatology trans-diagnostically with the aim of identifying constellations of related symptoms. Components were retained based on the eigenvalue one criterion.

### Imaging acquisition and pre-processing
All imaging data was acquired on 3 T Siemens Trio scanners. Each subject from both datasets had a structural T1 image and a resting state fMRI acquisition, although field maps were only available for CamCAN subjects. Extensive details on image acquisition and pre-processing can be found in the supplementary materials.

### Receptor-enriched Analysis of functional Connectivity by Targets (REACT)
We employed transporter and receptor density maps from the noradrenergic, dopaminergic, serotonergic, cholinergic, glutamatergic, and GABAergic systems. These are group average templates derived from healthy cohorts separate from the functional imaging datasets examined here. These have been widely utilised in our previous work[34–37,121–124], as well as by the broader imaging community[125,126]. Here, we chose to use transporters for the neuromodulatory systems as these provide a general measure of the innervation and influence of a given receptor system over a given region[33]. Further details regarding the templates are reported in the supplementary materials.

For each subject, voxel-wise functional networks associated with each of the molecular templates (NAT, DAT, SERT, VAChT, mGLuR5, and GABA-A) were estimated utilising a two-step multiple linear regression framework implemented in the REACT toolbox (version 0.1.7) (https://github.com/ottaviadipasquale/react-fmri[127]). In the first step, the molecular templates are spatially regressed against the fMRI data for each subject within a multiple linear regression, yielding time series which capture the dominant fluctuations for each molecular system[33]. These are then entered into a second multiple linear regression, where they are regressed against the BOLD time series from each voxel to produce a molecular-enriched network map for each molecular template. To limit computational burden, we then parcellated our voxel-wise networks using a custom combination of different atlases. For the cortex, we utilised the Schaefer 400 region parcellation[128]. We additionally added 15 regions from the Harvard-Oxford subcortical parcellation (left/right thalamus, left/right caudate, left/right putamen, left/right pallidum, left/right hippocampus, left/right amygdala, left/right accumbens, and brainstem)[129–132]. Finally, we also utilised 28 cerebellar grey matter regions (all vermal and hemispheric lobules) from the SUIT atlas[133–135]. Molecular-enriched FC values contained within each ROI were averaged, resulting in a total of 443 molecular-enriched network ROI values for each subject and each molecular system.

### Normative modelling
In essence, this approach is used to generate individual molecular-enriched FC deviation maps based on a model of what an individual's molecular-enriched network "should" look like, given their demographics. Previous experimental work has described various normative modelling approaches[24,27–31]. As in most prior work, we implemented our normative

models using the PCNtoolkit (version 0.27)[21]. We chose to utilise a hierarchical Bayesian regression (HBR) approach which can accommodate signal and noise variance in data from multiple sites by estimating different but connected mean and variance components through shared prior distributions across sites[25,136]. A set of HBR models[137] were trained on healthy participants data from both the UCLA and CamCAN datasets, utilising a 70/30 train test split (implemented with sklearn.model_selection.train_test_split to ensure each dataset is equally represented in both training and test data), resulting in CamCAN = 346/150 and UCLA = 78/33 train/test subdivisions. By including both the CamCAN and UCLA data, this allowed us to generate a relatively stable distribution of estimates across the healthy ageing lifespan. A separate HBR model was estimated for each ROI and each receptor system (i.e., a separate model is fit for each region of the brain within each of the different molecular-enriched networks derived from our REACT and parcellation pipeline), using the age and sex of participants to build a predictive model of molecular-enriched FC values. Specifically, we used site (UCLA, Cambridge) as a random effect as well as age and sex as fixed effects, resulting in normative regional molecular-enriched FC variance and uncertainty. Model performance was evaluated by the amount of variance explained when applied to the test sample. The subsequent analyses utilised only ROIs with positive explained variance (EV), in order to exclude areas where the model failed to converge on a stable estimate. Predictions using these models were also made for the clinical cohorts, where age and sex were used to predict their molecular-enriched networks. This provides both point estimates as well as measures of predictive confidence, allowing us to statistically quantify deviations from the normative molecular-enriched networks with regional specificity. Specifically, we computed a deviation score for each brain ROI, which describes the difference between the predicted molecular-enriched FC and the true molecular-enriched FC, normalised by the prediction uncertainty. In all subsequent analyses, we refer to the subset of healthy UCLA test subjects as $HC_{UCLA}$.

### Analysis of deviation scores between diagnostic groups

In line with the conventional diagnostic boundaries, we first examined whether the deviation scores differed between our three clinical cohorts and the $HC_{UCLA}$ group. This was implemented ROI-wise as a $1 \times 4$ ANOVA within FSL's Permutation Analysis of Linear Models programme (PALM[138] 2000 permutations and FDR correction). Lower-level t contrasts were used to delineate which groups were driving each significant higher-level result. To explore the added value of normative modelling, we repeated these analyses on the original molecular-enriched FC values for each network as would be done in a conventional REACT analysis (SI-figure 7).

We also collapsed down the deviation scores of each molecular-enriched network to provide a single mean deviation score for each subject across ROIs. This was motivated by the need for simple measures for putative biomarkers to be clinically practicable. We therefore used this summary metric describing whether FC of the network enriched with a given molecular system for a given subject is globally shifted more towards positive or negative deviations from normality ranges as well as the magnitude of this shift. We then conducted a $1 \times 4$ ANOVA for each molecular system to see whether these summary metrics differed between our three clinical cohorts and the $HC_{UCLA}$ subjects. This analysis was implemented in Python (SciPy stats and statsmodels) with Tukey correction for lower-level comparisons. To further investigate the diagnostic capabilities of these summary metrics, we took significant between-group differences from $HC_{UCLA}$ at the lower level and entered these mean values into a binary logistic regression analysis. Receiver operating characteristic curves (ROC) were plotted to display the trade-off between sensitivity and specificity. Area under the curve (AUC) was used to quantify predictive performance. This was also implemented in Python using statsmodels and sklearn.

### Analysis of between-subject FC deviation similarity

We considered a novel approach to quantify how similar patterns of deviations were within and between groups. We speculated that whilst the

$HC_{UCLA}$ individuals would show a random pattern of deviations from normality, deviations within the patient groups may converge on similar patterns relating to a common disease process(es). We also expected the groups of patients to be more similar to one another than $HC_{UCLA}$, signifying transdiagnostic similarity. Correlation matrices were generated for each molecular system, with each showing the correlation of FC deviation scores across all brain regions between each pair of individuals from the UCLA dataset ($HC_{UCLA}$, SCHZ, BPD, and ADHD). This provides an overview of similarity across individuals, including both how similar the pattern of deviations is to those within the same group (within-group similarity) as well as to those in a different group (between-group similarity). We first compared the distributions of the within-group correlation coefficients across datasets using two-sample Kolmogorov-Smirnov tests (implemented in Python with scipy.stats). Additionally, to assess the extent of similarity both within and across groups, we determined for each patient their average similarity to individuals of their own group (within-group similarity) and individuals from the other two groups (between-group similarity). Subsequently, analogous to what was done with psychometric data, we employed non-parametric repeated measures ANOVAs using the Friedman test, along with Conover's post-hoc analysis, to investigate differences in within- and between-group similarity among the groups, applying Bonferroni correction for multiple comparisons.

Finally, we estimated a transdiagnostic similarity metric to quantify the average similarity of each patient to all other individuals across clinical groups. This summary measure was then correlated with the summary metric of mean FC deviation to examine the presence of a linear relationship between FC deviation similarity and its magnitude on a transdiagnostic level. Furthermore, we investigated the link between this similarity in FC deviations and clinical sub-domains identified through PCA. This was based on the speculation that individuals with closely matching FC deviations from normative ranges might share a more uniform pathological brain state, as indicated by their symptom profiles. This analysis was implemented in Python and Bonferroni corrected for multiple comparisons across components and molecular systems ($p < 0.05/30$).

### Transdiagnostic deviation-symptom mapping

The relationships between the four symptom-based PCs and the molecular-enriched network deviations from normality, expressed through the deviation scores, were examined through mass univariate regression analyses. The resulting correlation coefficients reflect the magnitude of the statistical relationship between clinical and deviation data within each ROI across the brain. This was implemented through non-parametric permutation testing (2000 permutations and FDR correction) using PALM. In order to examine to what extent the normative modelling approach provides additional benefit, we also ran the exact same analysis on the raw molecular-enriched FC values, as would be done within a conventional REACT analysis (SI-figure 7).

### Data availability

All data is available from open repositories as outlined within section 6.1. Specifically, two datasets were used: the CamCAN repository available at http://www.mrc-cbu.cam.ac.uk/datasets/CamCAN/ and the UCLA phenomics dataset available from https://openneuro.org/datasets/ds000030/versions/1.0.0).

### Code availability

This work uses openly available and widely used pipelines and toolboxes, with scripts implementing these available upon request.

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

## Acknowledgements

T.L. is in receipt of a PhD studentship funded by the National Institute for Health Research (NIHR) Biomedical Research Centre at South London and Maudsley NHS Foundation Trust and King's College London. DM, MAH, and FET are supported by the NIHR Biomedical Research Centre and Clinical Research Facility at South London and Maudsley NHS Foundation Trust and King's College London. MAH is also supported by the Medical Research Council (MR/N026969/1). M.V. is supported by EU funding within the MUR PNRR "National Centre for HPC, BIG DATA AND QUANTUM COMPUTING (Project no. CN00000013 CN1), by the PNR National Grant DIGITAL LIFELONG PREVENTION (Project no PNC0000002_DARE), and by Wellcome Trust Digital Award (no. 215747/Z/19/Z). A.G. is supported by the KCL-funded CDT in Data-Driven Health; this represents independent research partly funded by the NIHR Maudsley's Biomedical Research Centre (BRC) at the South London and Maudsley NHS Foundation Trust and partly funded by GSK. The views expressed are those of the authors and not necessarily those of the NHS, the NIHR, or the Department of Health and Social Care.

## Competing interests

The authors declare no competing interests.
