## [Peer review file · Communications Biology]

Reviewers' comments:

Reviewer #1 (Remarks to the Author):

In "Normative modeling of molecular-based functional circuits captures clinical heterogeneity transdiagnostically in psychiatric patients" by Lawn et al, the authors construct normative models of "molecular-based functional circuits" (derived from PET-estimated receptor densities and BOLD fMRI time-series) and use this normative model to assess patient deviation from the norm, across three psychiatric diagnostic categories (schizophrenia, bipolar disorder, ADHD). The authors emphasize the novelty of the work being an integration of (1) normative modeling (and therefore avoiding group-averages but rather being able to analyze subject-specific data, which is especially important for clinical studies) and (2) multiple descriptions of brain organization, that is, molecular mechanisms (receptors, from PET) and functional dynamics (BOLD signal, from fMRI). Overall, I congratulate the authors for putting together many different data types to study psychiatric disorders at the level of the individual. I did find the methods a bit difficult to follow, so I have some suggestions for improvement prior to publication.

1. I found it difficult to follow all the methodology and keep track of all the models. A schematic figure illustrating the methods would be helpful.
2. (Related to the point above) perhaps also adding some "take away" sentences to the subsections in the Results would help, eg "Collectively, we show that ..." or "this demonstrates that ...".
3. Since REACT is an important part of the analyses, the authors should explain the methods for REACT in detail rather than pointing readers to a separate paper. I recommend a brief description in the Results, a complete description in the Methods, and also to expand on Supplementary Figure 1a caption, which currently does not describe all that is in Figure S1a.
4. Throughout the manuscript, the authors emphasize the fact that diagnostic categories are poorly defined for many psychiatric conditions. Indeed, psychometric measurements don't clearly define the three groups (Figure 1). However, the authors also stick to these pre-defined diagnostic categories throughout their study. Wouldn't it be relevant to see whether the normative modeling can identify individuals that "deviate" similarly, above and beyond diagnostic category? Likewise, could the authors use the dimensionality reduction to identify psychometric measurements that better classify individuals with respect to how their receptor-informed functional networks deviate from the norm? My main point is that I felt that the authors emphasized the limitations of these diagnostic categories a lot only to then use them to assess their normative model, which seemed a bit contradictory.

5. Could the authors show the average z brain maps for each diagnostic category and each molecular system?

6. If I understand correctly, the authors are using “hierarchical organization of the brain” to refer to the fact that the brain is a complex system that integrates multiple levels of description (eg genes, receptors, cells, all the way to macroscale dynamics, functional networks, behaviour etc). The term “hierarchy” can be interpreted in many ways (eg see Hilgetag & Goulas 2020 <https://royalsocietypublishing.org/doi/10.1098/rstb.2019.0319>), for example, at first I thought the authors were referring to the functional hierarchical spatial organization of the brain. I therefore recommend clarifying and/or removing the language of “hierarchy”.

7. Figure 2: there is a mismatch between the values reported in Figure 2b piechart (57.3%) and the values reported in the text (68.9%).

8. More generally, the text in all the figures was very small and difficult to read.

9. In the Discussion the authors emphasize the finding about “implicat[ing] the excitatory-inhibitory balance in SCHZ and BPD”. I don’t follow how the authors reached this conclusion - perhaps some clarification in the Results would be helpful.

Minor:

10. Typo in the Abstract: “REACT was used [to] create ...”

11. Abstract: “... in order to address these two longstanding translational barriers”. At this point in the abstract, it wasn’t clear to me what the two “longstanding translational barriers” were (this is later clarified in the Introduction). I would recommend listing out the two barriers in clear language here, or removing the statement.

12. Introduction, paragraph #1: “... components interact across scales and time to bring...”; “scales” can refer to many things - I would rephrase as “across spatial and temporal scales”, or something similar.

13. Introduction, paragraph #3: “Neuroimaging offers a non-invasive set of methods which can measure the structure and function of the brain” - perhaps I am being too picky but I thought this language was very strong. Maybe rephrase as “which facilitates the study of brain structure and function”?

14. Results section 2.6 paragraph #1: Figure subpanel labels are mismatched (3B, 3C instead of 3A, 3B).

15. Results section 2.7 paragraph #1, towards the end: "... we found significant negative correlations [for?] mGluR5 and GABA-A following Bonferroni correction...". Something weird about this sentence, maybe the missing "for" but please double check.

16. Discussion section 3.1, please cite at least some of the studies "published previously" that are being referred to in the first sentence of this section.

17. Discussion section 3.4, there is a stray "t" in the middle of the paragraph ("Future methodological development will be crucial to t carefully tease apart...").

18. Methods section 6.6, not sure what was meant by "Specifically, these model site as a random effect and age and sex as fixed effects" - I think there is a typo/missing word somewhere.

Reviewer #2 (Remarks to the Author):

This is a very interesting paper combining two novel and powerful analytical approaches to characterize functional brain alterations across a range of psychiatric conditions. Some minor but important concerns need to be addressed before it is suitable for publications.

Introduction:

1) Very well written. I would remove the unnecessary summary of the results/conclusion that is currently featured at the end of the introduction. It would instead be beneficial to have a more extensive intro on REACT and in particular on the biological meaning of the REACT-derived networks and in particular of the positive/negative deviations that are mentioned later on in the results. This would make results easier to understand for readers unfamiliar with the REACT-approach

2) Several neurotransmitters templates exist and are freely available (30+ e.g. in the neuromaps toolbox). Why did the authors selected some neurotransmission systems, specifically and why did they select the specific targets for each neurotransmission system (e.g. instead of DAT one could have targeted D1, D2 receptors etc)? Is this due to the way the REACT analysis works, with all molecular information entering together in the dual regression?

Most targets selected are presynaptic and only one is post-synaptic. How does this impact the REACT analysis that rely on BOLD signal which is mainly reflective of post-synaptic potentials?

Methods

3) PCA results are somewhat dependent on the parameters specified by the experiment. To increase transparency and possibility to reproduce results shown by the authors, it would be beneficial to report the parameters applied in the PCA. For example, what rotation was applied, if any? Why the authors chose one type of rotation over another? How does this impact the number and characteristics of the identified PCs?

4) The authors selected ROIs from different atlases, some of them having very fine resolution (cortical regions) and other being extremely coarse (brainstem). How does this affect results of REACT? Is selection of ROIs with fine resolution appropriate with PET and in particular SPECT images that have poorer resolution than fMRI? Normally, for PET, it is suggested to select ROIs with twice the volume of the FWHM

Results

5) Sample: Were the patients drug-naive and/or drug-free at the time of acquisition? This is a very important point that should be highlighted and discussed.

6) Section 2.2: this section is very descriptive; it would be beneficial if authors could provide some actual stats instead of reporting that "that there are clear examples of.."

7) Here and throughout the paper the authors refer to "ROIs for each molecular system". I would recommend opting for a different wording that is mindful of the data used in the paper, e.g. referring to "molecularly-enriched functional systems" or the like

8) It is not clear to me why the authors evaluate whether PCs are related to the level of within-group similarity. Could you please explain? I do not see any biological relation between the two things. Is there a statistical/technical reason instead?

9) It would be helpful if the authors could define how to interpret negative vs positive Z scores, from a biological point of view (section 2.8). This is not straightforward for a reader unfamiliar with REACT.

10) It would help the reader to collect all sensitivity analyses run without normative modelling in a separate paragraph at the end of the result section and/or in supplementary materials.

Discussion

11) The authors state that their model can explain a comparable amount of variance of models published previously. This is a bit surprising - don't the authors expect their model, including molecular information on top of functional ones, to explain a higher amount of variance than standard normative models? Please discuss. In general paragraph 3.1 is rather general and feels like a repetition of the introduction and conclusions

12) "negative deviations" (3.2), please explain

13) "although some of these measures have also shown null findings..". It would be beneficial to mention here a meta-analysis, if any exists, showing that the literature overall indeed points towards the authors' mention of a significant effect

14) Why was a D2-enriched network not investigated (on top of or instead of DAT) if D2 treatments are so commonly used in SHZ?

15) "the lack of results for ADHD throughout our analysis...". Do the author's clinical and neuroimaging findings confirm that indeed ADHD is more heterogeneous than the other psychiatric cohorts included in this study? I wonder if, aside of possibly bigger heterogeneity, the lack of significant results is simply due to smaller effects of ADHD on brain activity as compared to more dramatic/pervasive conditions such as SCHZ and BPD

16) Limitations and throughout the manuscript: I'd suggest to refer to molecular imaging and/or PET/SPECT instead of PET templates only (incorrect). Differences in spatial resolution between templates (PET/SPECT and among different PET scanners, if any) should also be acknowledged in this section.

Supplementary Materials

17) Fig.1. It would be beneficial to render the PET and SPECT templates (simple axial render) used by the author prior to parcellation to let the reader appreciate the eventual differences in spatial resolution between templates.

18) Table 2. It would helpful to report in the table the number of subjects and of ROIs included in this analysis

19) Related to point 2) and at consistence with the rationale of the paper, it would be relevant to report the mean and range of age and sex of the control subjects based on which the neurotransmission templates were obtained. The authors should also justify the choice of a SPECT template over a PET one for the dopaminergic system (worse spatial resolution) and of templates based on very limited number of subjects (e.g. for GABA-A why not use beliveau2017_dasb_MNI152_1mm based on 16 subjects?)

General:

20) References are sometimes reported in an inconsistent format in main text and sm

21) Typo in the abstract

Response to reviewers document

We thank both reviewers for their constructive and insightful comments on our manuscript. Here, we intersperse our response to each point in turn using **red text**, referring to associated changes made within the main manuscript document and supplementary materials, which we highlight in **blue text** for your convenience. The resulting work is undoubtedly much improved thanks to your thoughtful input.

Reviewer #1 (Remarks to the Author)

In “Normative modelling of molecular-based functional circuits captures clinical heterogeneity transdiagnostically in psychiatric patients” by Lawn et al, the authors construct normative models of “molecular-based functional circuits” (derived from PET-estimated receptor densities and BOLD fMRI time-series) and use this normative model to assess patient deviation from the norm, across three psychiatric diagnostic categories (schizophrenia, bipolar disorder, ADHD). The authors emphasize the novelty of the work being an integration of (1) normative modelling (and therefore avoiding group-averages but rather being able to analyse subject-specific data, which is especially important for clinical studies) and (2) multiple descriptions of brain organization, that is, molecular mechanisms (receptors, from PET) and functional dynamics (BOLD signal, from fMRI). Overall, I congratulate the authors for putting together many different data types to study psychiatric disorders at the level of the individual. I did find the methods a bit difficult to follow, so I have some suggestions for improvement prior to publication.

1. I found it difficult to follow all the methodology and keep track of all the models. A schematic figure illustrating the methods would be helpful.

We appreciate the complexity of the methods here and agree that a schematic providing an overview is worthwhile. We have included this within the manuscript.

2. (Related to the point above) perhaps also adding some “take away” sentences to the subsections in the Results would help, eg “Collectively, we show that ...” or “this demonstrates that ...”.

We have gone through the results section and tried to further add some take away messages, whilst trying to be mindful of not providing interpretation of the results, as this is done later on within the discussion.

3. Since REACT is an important part of the analyses, the authors should explain the methods for REACT in detail rather than pointing readers to a separate paper. I recommend a brief description in the Results, a complete description in the Methods, and also to expand on Supplementary Figure 1a caption, which currently does not describe all that is in Figure S1a.

We have implemented all of these suggestions. Specifically:

- We provide some further clarification within results to further orient the reader to what this method provides.
- REACT is now fully described in the methods.
- We add additional information into the legend of SI-figure 1 such that it offers a more substantive standalone explanation.

4. Throughout the manuscript, the authors emphasize the fact that diagnostic categories are poorly defined for many psychiatric conditions. Indeed, psychometric measurements don't clearly define the three groups (Figure 1). However, the authors also stick to these pre-defined diagnostic categories throughout their study. Wouldn't it be relevant to see whether the normative modelling can identify individuals that "deviate" similarly, above and beyond diagnostic category? Likewise, could the authors use the dimensionality reduction to identify psychometric measurements that better classify individuals with respect to how their receptor-informed functional networks deviate from the norm? My main point is that I felt that the authors emphasized the limitations of these diagnostic categories a lot only to then use them to assess their normative model, which seemed a bit contradictory.

Thank you for highlighting this crucial aspect. Our findings relate both to traditional diagnostic criteria, as evidenced by the ANOVA results in section 2.6, and progressively adopt a transdiagnostic perspective. Following the invaluable feedback from reviewers, we have clarified this transition in the revised manuscript. In sections 2.2, 2.6, and 2.7, we assess how well diagnostic criteria reflect the underlying patterns of alterations across patients grouped by the same diagnostic label, in terms of both symptoms and FC deviations from normative ranges. Although similarities between patients with the same diagnosis are evident, these criteria alone are insufficient for a comprehensive understanding of the disorders' pathophysiology. For instance, patients with schizophrenia exhibited greater similarity in symptomatology and molecular-enriched FC deviations compared to the other two groups. These observations suggest that discarding diagnostic labels in favour of exclusively transdiagnostic analyses may not capture the disorders in full. However, our exploration beyond traditional categories, particularly in symptom sub-domains within sections 2.8 and 2.9, uncovers significant correlations between FC deviations and pathophysiology. Our approach aims to balance both perspectives, advocating for a shift towards mechanisms-based methodologies.

Regarding the possibility of redefining subgroups through a component reduction of psychometric measures, our approach in current section 2.9, i.e. the transdiagnostic deviation-symptom mapping, precisely attempted this, although in a more straightforward way without the application of machine learning algorithms for classification. While figure 6B visually represents patients with the colours corresponding to their diagnostic groups, our mass univariate regression analysis was conducted transdiagnostically, aiming to identify correlations between symptomatology and FC deviations without considering the diagnostic labels. We found that deviations in the VAcHT- and mGluR5-enriched networks are associated with depressive and psychotic symptomatology captured by PC2. Future research requiring larger sample sizes will be necessary to corroborate these findings, as this

investigation goes beyond the scope of the current project. Nevertheless, our results offer a preliminary validation of this concept, warranting further exploration and confirmation in follow up work.

5. Could the authors show the average z brain maps for each diagnostic category and each molecular system?

Yes, we have added a figure showing the average deviation maps, expressed in z-score, for each diagnostic category (SCHZ, BPD, ADHD), and receptor system to the supplementary materials. This is now supplementary figure 3.

6. If I understand correctly, the authors are using “hierarchical organization of the brain” to refer to the fact that the brain is a complex system that integrates multiple levels of description (eg genes, receptors, cells, all the way to macroscale dynamics, functional networks, behaviour etc). The term “hierarchy” can be interpreted in many ways (eg see Hilgetag & Goulas 2020 <https://royalsocietypublishing.org/doi/10.1098/rstb.2019.0319>), for example, at first I thought the authors were referring to the functional hierarchical spatial organization of the brain. I therefore recommend clarifying and/or removing the language of “hierarchy”.

Thank you for pointing this out, this term might be ambiguous and potentially misleading, so we have replaced it throughout the manuscript, generally opting instead for “multiscale”.

7. Figure 2: there is a mismatch between the values reported in Figure 2b pie chart (57.3%) and the values reported in the text (68.9%).

Many thanks for spotting this – we have amended the value in the legend to match the correct value in the figure.

8. More generally, the text in all the figures was very small and difficult to read.

Thank you for pointing this out. We have gone back to the figures and tried to improve their readability.

9. In the Discussion the authors emphasize the finding about “implicating the excitatory-inhibitory balance in SCHZ and BPD”. I don’t follow how the authors reached this conclusion perhaps some clarification in the Results would be helpful.

We now attempt to clarify this, changing this sentence to now read “Our between-group findings converge to broadly implicate the glutamate and GABA in SCHZ and BPD (possibly alluding to altered excitatory-inhibitory balance)...”, which hopefully makes this somewhat clearer before it is explained further within 3.2.

Minor:

10. Typo in the Abstract: “REACT was used [to] create ...”

Thank you for spotting this; the missing “to” has been added.

11. Abstract: "... in order to address these two longstanding translational barriers". At this point in the abstract, it wasn't clear to me what the two "longstanding translational barriers" were (this is later clarified in the Introduction). I would recommend listing out the two barriers in clear language here, or removing the statement. This was intended to refer to the previous statement "This has been hindered by the multiscale organisation of the brain and heterogeneity of psychiatric disorders". We have now reworked the abstract to try and make this unambiguously clear.

12. Introduction, paragraph #1: "... components interact across scales and time to bring..."; "scales" can refer to many things- I would rephrase as "across spatial and temporal scales", or something similar.

Thank you for this, we have replaced it as suggested.

13. Introduction, paragraph #3: "Neuroimaging offers a non-invasive set of methods which can measure the structure and function of the brain"- perhaps I am being too picky but I thought this language was very strong. Maybe rephrase as "which facilitates the study of brain structure and function"?

Thank you for the advice, this has been amended according to your suggestion.

14. Results section 2.6 paragraph #1: Figure subpanel labels are mismatched (3B, 3C instead of 3A, 3B).

Thank you for pointing this out; we have amended these labels.

15. Results section 2.7 paragraph #1, towards the end: "... we found significant negative correlations [for?] mGluR5 and GABA-A following Bonferroni correction...". Something weird about this sentence, maybe the missing "for" but please double check.

Yes, you are correct that this sentence was missing a "for" which has now been added.

16. Discussion section 3.1, please cite at least some of the studies "published previously" that are being referred to in the first sentence of this section.

Thank you for pointing this out. We have added the citation of a recent paper which gives clear reporting of explained variance from voxel-based morphometry within a similar normative modelling analysis.

17. Discussion section 3.4, there is a stray "t" in the middle of the paragraph ("Future methodological development will be crucial to t carefully tease apart...").

Thank you for catching this typo; it has been amended.

18. Methods section 6.6, not sure what was meant by "Specifically, these model site as a random effect and age and sex as fixed effects"- I think there is a typo/missing word somewhere.

Thanks for pointing this out. We have changed this to read "Specifically, we used site (UCLA, Cambridge) as a random effect as well as age and sex as fixed effects [...]", which is hopefully clearer.

Reviewer #2 (Remarks to the Author):

This is a very interesting paper combining two novel and powerful analytical approaches to characterize functional brain alterations across a range of psychiatric conditions. Some minor but important concerns need to be addressed before it is suitable for publications.

Introduction:

1) Very well written. I would remove the unnecessary summary of the results/conclusion that is currently featured at the end of the introduction. It would instead be beneficial to have a more extensive intro on REACT and in particular on the biological meaning of the REACT-derived networks and in particular of the positive/negative deviations that are mentioned later on in the results. This would make results easier to understand for readers unfamiliar with the REACT-approach.

As suggested, we have removed comments on the results and add in additional information regarding REACT in the introduction and methods. We have also added some details to the normative modelling paragraph and explained what positive and negative deviations from normality are, in order to introduce this concept earlier within the manuscript. We also clarified this further later on, adding the following into the results to help orient readers further to what we mean by this: "These deviations can be either positive or negative values reflecting the degree to which the subject's molecular-enriched FC value for that brain region is higher or lower than the normal population ranges, respectively". To provide a clear message to the readers, we have also rephrased the parts where we mentioned these "more positive and more negative deviations", using the wording "molecular-enriched FC greater or lower than the normal population ranges". We hope that this completely disambiguates these terms prior to their use within the remaining results section.

2) Several neurotransmitters templates exist and are freely available (30+ e.g. in the neuromaps toolbox). Why did the authors selected some neurotransmission systems, specifically and why did they select the specific targets for each neurotransmission system (e.g. instead of DAT one could have targeted D1, D2 receptors etc)? Is this due to the way the REACT analysis works, with all molecular information entering together in the dual regression? Most targets selected are presynaptic and only one is post-synaptic. How does this impact the REACT analysis that rely on BOLD signal which is mainly reflective of post-synaptic potentials?

These are very important considerations and thanks for giving us the opportunity to clarify these issues. We go into detail on these points within our recent review paper – <https://www.sciencedirect.com/science/article/pii/S0149763423001628>. To summarise:

Selection of neurotransmitter systems for templates

- This does largely relate to the fact that the molecular information is included together within a multiple-regression framework. We have used the transporters for different systems in previous work as well as the current manuscript. As we state in 6.5, we chose to use transporters for the neuromodulatory systems

as these provide a general measure of the innervation and influence of a given receptor system over a given region.

- Whilst far from identical, the DAT and D2 maps capture similar information given their relatively strong colocalization, which translates into a collinearity issue when using both into the same linear model. We have generally found the transporters to offer a useful approximation for a given molecular system's distribution throughout the brain. Furthermore, by using transporters, we can examine the systems from a more general perspective, while future work could build upon this to examine specific receptor sub-systems within suitable datasets. For example, a dataset which has information as to treatment response might want to be more specific in choosing specific receptor targets engaged by that treatment.

In essence, by taking the transporters of the main modulatory neurotransmitters as well as GABA and glutamate receptors, we covered a substantial amount of the underlying neurobiology within a small number of templates, which we felt offered a solid foundation for this ambitious and exploratory analysis, without running afoul of including a large number of molecular templates in a single model.

Pre- and post-synaptic localisation

This is a very pertinent question for which we do not have a definitive answer. The modulatory neurotransmitter systems project from small brainstem and midbrain nuclei to disparate parts of the brain. As such, their influence on the BOLD signal and network dynamics is likely well captured by the location of the synapse. Additionally, given the spatial resolution of 2mm^3 , it likely makes little difference whether the target of the radioligand is pre or post synaptic; it seems probable that there is good colocalization between the receptor density and the BOLD signal. Whilst we are not aware of any definitive evidence of this, a large body of work (for example <https://www.nature.com/articles/s41593-022-01186-3>) has demonstrated robust relationships between PET/SPECT-derived molecular density templates and functional imaging measures, providing additional confidence in this idea.

Methods

3) PCA results are somewhat dependent on the parameters specified by the experiment. To increase transparency and possibility to reproduce results shown by the authors, it would be beneficial to report the parameters applied in the PCA. For example, what rotation was applied, if any? Why the authors chose one type of rotation over another? How does this impact the number and characteristics of the identified PCs?

As now clarified in the manuscript, we opted for the "variance maximizing" rotation, which is essentially the orthogonal (or no rotation) method. The decision to refrain from rotation was based on a few considerations.

- The orthogonal rotation method was chosen in order to obtain uncorrelated and interpretable principal components. In our case, our primary aim was to identify underlying factors in the psychometric data without assuming a specific pattern of relationships between the original variables. This allows for a clearer and more direct interpretation of the resulting principal components.

- Orthogonal rotation simplifies the interpretation of principal components by avoiding the introduction of correlations between them, providing a meaningful representation of the underlying structure without assuming complex interrelationships. The absence of rotation ensures that the identified principal components are straightforward linear combinations of the original variables, which offers simplicity.

While we acknowledge that the choice of rotation method can impact the outcome of PCA, our rationale for selecting the orthogonal rotation method was rooted in the specific characteristics of our data and the desire to present results in a clear and easily interpretable manner.

4) The authors selected ROIs from different atlases, some of them having very fine resolution (cortical regions) and other being extremely coarse (brainstem). How does this affect results of REACT? Is selection of ROIs with fine resolution appropriate with PET and in particular SPECT images that have poorer resolution than fMRI? Normally, for PET, it is suggested to select ROIs with twice the volume of the FWHM

Thanks for mentioning this, we realised that we weren't sufficiently clear about this point. We do not parcellate the PET or SPECT templates at any stage. These are used voxel-wise as regressors within REACT, as now specified: *"For each subject, voxel-wise functional networks associated with each of the molecular systems (NAT, DAT, SERT, VACHT, mGluR5, and GABA-A) were estimated utilising a two-step multiple linear regression framework implemented in the REACT toolbox"*. We then subsequently parcellated the resultant molecular-enriched networks, as now reported next *"In order to limit computational burden, we then parcellated our voxel-wise networks using a custom combination of different atlases"*. The choice of ROIs generally reflects our attempts to balance spatial resolution with computational expense of our hierarchical Bayesian regression normative models. We found that the total number of ROIs (443) achieved by combining these atlases met this balance well. We fully acknowledge that the brainstem ROI is very coarse, and in fact that most parcellations would not include any brainstem regions. Here, we feel that this likely made negligible difference to any of our analyses.

Results

5) Sample: Were the patients drug-naïve and/or drug-free at the time of acquisition? This is a very important point that should be highlighted and discussed.

Patients were not drug naïve nor drug free. We fully acknowledge that this is a very important consideration, so it has been discussed as a limitation in the discussion: *"Finally, we cannot exclude the possibility that treatment within the clinical cohorts may impact the estimation of deviations within the molecular-enriched functional networks, potentially reducing the extent of deviations in those subjects responding well to treatment while inducing deviations in regions of high target engagement but minimal contribution to pathology. Future studies examining drug-naïve populations, or with sample sizes sufficiently large to attempt to control for treatment type, will be important moving forwards."*

6) Section 2.2: this section is very descriptive; it would be beneficial if authors could provide some actual stats instead of reporting that 'that there are clear examples of..'

We have modified this section to try make it less descriptive and more quantitative. Specifically, we now move the density curves to the supplement and undertake additional quantification of within- and between-group similarity metrics which we statistically compare.

7) Here and throughout the paper the authors refer to "ROIs for each molecular system". I would recommend opting for a different wording that is mindful of the data used in the paper, e.g. referring to "molecularly-enriched functional systems" or the like

We have made the suggested modification throughout the manuscript.

8) It is not clear to me why the authors evaluate whether PCs are related to the level of within-group similarity. Could you please explain? I do not see any biological relation between the two things. Is there a statistical/technical reason instead?

Many thanks for pointing this out. Upon further consideration, we agree that this was not the best way to approach this analysis and we have now made a significant change. Instead of using within-group similarity to relate through to the composite symptom scores, we derive "transdiagnostic similarity", which instead captures the average similarity of each patient to every other patient across groups. This therefore provides an interesting metric of how much the similarity patterns of deviations map onto each symptom sub-domain, irrespective of the diagnostic domains. Interestingly, we find a similar pattern of results using this approach, indicating that PC2 is related to transdiagnostic similarity within the serotonergic, cholinergic, and glutamatergic systems after stringent Bonferroni correction. We have incorporated this into the manuscript and modified the results and discussion accordingly.

9) It would be helpful if the authors could define how to interpret negative vs positive Z scores, from a biological point of view (section 2.8). This is not straightforward for a reader unfamiliar with REACT.

Thank you for pointing this out. The positive and negative deviations have been also reported throughout the manuscript as z scores, although what we mean is that the deviations are expressed as z-scores. Since this might be confusing, we decided to use the wording "deviations" everywhere. From a biological point of view, since FC of brain regions can assume either positive or negative values, depending on their connectivity with the network under exam, positive and negative deviations will indicate abnormal greater/smaller FC as compared to the normal population. This should help disambiguate what we mean by positive/negative deviations from the perspective of functional connectivity, and has now been clarified in the manuscript.

10) It would help the reader to collect all sensitivity analyses run without normative modelling in a separate paragraph at the end of the result section and/or in supplementary materials.

As suggested, these sensitivity analyses have been collected into a new subsection and moved into the supplementary materials.

Discussion

11) The authors state that their model can explain a comparable amount of variance of models published previously. This is a bit surprising- don't the authors expect their model, including molecular information on top of functional ones, to explain a higher amount of variance than standard normative models? Please discuss. In general paragraph 3.1 is rather general and feels like a repetition of the introduction and conclusions

Thanks for allowing us to clarify this important point. So far, the predominant focus in normative modelling has been on structural data, largely due to its interpretability and intrinsic lower variability of that kind of data. Functional imaging data is higher dimensional and noisier than structural data, which translates into a greater variability of this type of measure as compared to structural measures such as grey matter volume and cortical thickness. For this reason, we expected that our functional networks would be harder to predict from demographic variables, so obtaining comparable EV ranges to prior work was reassuring. Indeed, our results encourage a broader adoption of this approach using molecular-enriched networks, as it may offer substantial benefits as biomarkers for precision medicine whilst offering comparable model performance. We have tried to clarify this point within section 3.1.

12) "negative deviations" (3.2), please explain

Thanks for pointing out that we didn't specifically clarify what we meant by this. Negative deviations refer to the directionality of deviation from the models. In other words, a negative deviation reflects a FC value that is lower than expected from what the model predicted as normal range, whilst a positive deviation Z score reflects a value that is greater than expected. As discussed in response to above comments, we have added explanations to the manuscript to help the reader better follow this terminology. We have also added a clarification in 3.2.

13) "although some of these measures have also shown null findings..". It would be beneficial to mention here a meta-analysis, if any exists, showing that the literature overall indeed points towards the authors' mention of a significant effect

This statement refers to very diverse lines of evidence spanning genetics through to behaviour, and so sadly no meta-analysis can directly support this statement. However, instead we offer several citations of review papers which broadly overview this evidence and narratively support this statement. Hopefully this provides sufficient clarification.

14) Why was a D2-enriched network not investigated (on top of or instead of DAT) if D2 treatments are so commonly used in SHZ?

This is essentially subsumed under our response to the query from the introduction section (comment 2). However, as mentioned above, whilst far from identical, the DAT and D2 maps capture similar information given their relatively strong colocalization, which translates into a collinearity issue when using both into the same linear model. We have generally found the transporters to offer a useful approximation for a given molecular system's distribution throughout the brain. Furthermore, by using transporters, we can examine the systems from a more general perspective, while future work could build upon this to examine specific receptor sub-systems within suitable datasets. For example, a dataset which has information as to treatment response might want to be more

specific in choosing specific receptor targets engaged by that treatment. This would therefore be better suited to an analysis of only schizophrenia patients within a dataset which provides outcome measures for antipsychotic responsiveness.

15) "the lack of results for ADHD throughout our analysis...". Do the author's clinical and neuroimaging findings confirm that indeed ADHD is more heterogeneous than the other psychiatric cohorts included in this study? I wonder if, aside of possibly bigger heterogeneity, the lack of significant results is simply due to smaller effects of ADHD on brain activity as compared to more dramatic/pervasive conditions such as SCHZ and BPD.

Thanks for raising this interesting argument. We acknowledge that the absence of significant findings for ADHD in our analysis might suggest not only potentially greater heterogeneity within ADHD compared to other psychiatric conditions like SCHZ and BPD, but also the possibility that ADHD impacts brain activity to a lesser extent, resulting in more nuanced effects. This interpretation is consistent with broader discussions in the scientific community. However, the notion of ADHD's heterogeneity is also well-documented in the literature, not just as an inherent feature of the disorder but also as a critical factor influencing methodological choices, such as the adoption of normative modelling approaches. For example, as we discuss in section 3.2, a meta-analysis involving 96 studies found inconsistent results, highlighting the variability within ADHD research, even though individual studies did report specific findings for their respective cohorts. This evidence underscores ADHD's significant individual and sample variability. Moreover, we acknowledge the potential for more subtle pathophysiological changes in ADHD that may be less detectable, further complicating the interpretation of these findings. In the amended version of the manuscript, we have also considered the possibility that ADHD's pathophysiological alterations might be more subtle and, consequently, more challenging to detect, adding another layer to our understanding of these results.

16) Limitations and throughout the manuscript: I'd suggest to refer to molecular imaging and/or PET/SPECT instead of PET templates only (incorrect). Differences in spatial resolution between templates (PET/SPECT and among different PET scanners, if any) should also be acknowledged in this section.

This is an oversight on our part and we have changed this to state PET/SPECT upon first use where we then state we will refer to these as molecular templates subsequently.

Supplementary Materials

17) Fig.1. It would be beneficial to render the PET and SPECT templates (simple axial render) used by the author prior to parcellation to let the reader appreciate the eventual differences in spatial resolution between templates. As stated in a previous point, molecular templates did not go through subsequent parcellation. However, we realised that we did not report in the "Population-based molecular templates" section that before doing the described processing of the templates (i.e., minimisation of the contribution of the reference regions and normalisation between 0 and 1), we resampled each template to 2 mm³. This is a preliminary mandatory step that allows to use all templates in the same GLM of the first part of REACT. Thanks for pointing this out, it has been amended. We have added a new figure (supplementary figure 3) which shows these, as well as average deviation maps for each group.

18) Table 2. It would helpful to report in the table the number of subjects and of ROIs included in this analysis
These have both been added to the figure legend.

19) Related to point 2) and at consistence with the rationale of the paper, it would be relevant to report the mean and range of age and sex of the control subjects based on which the neurotransmission templates were obtained. The authors should also justify the choice of a SPECT template over a PET one for the dopaminergic system (worse spatial resolution) and of templates based on very limited number of subjects (e.g. for GABA-A why not use believeau2017_dasb_MNI152_1mm based on 16 subjects?)

We have added the details of age and sex into the section of the supplementary materials entitled “Population-based molecular templates”. However, this was only possible where this information was reported in the original publications. The selection of which templates to use requires balancing many factors. In general, we have selected templates that will work well together. To take the example of GABA-A, the template you reference here is very high resolution and cortical regions are defined using surface-based algorithms. To be consistent with the other molecular maps, we decided to opt for a lower resolution but wider coverage of the grey matter.

General:

20) References are sometimes reported in an inconsistent format in main text and sm

Thank you for catching these – this was due to a formatting error. They have all been amended.

21) Typo in the abstract

Thank you for spotting this; the missing “to” has been added.

REVIEWERS' COMMENTS:

Reviewer #1 (Remarks to the Author):

The authors have addressed all of my comments and I endorse the publication of this manuscript.

Reviewer #2 (Remarks to the Author):

The authors addressed most of my comments. However, there is still some clarifications that I'd appreciate seeing implemented. Numbers here refer to the number of the comments in the first revision round

1) "These deviations can be either positive or negative values reflecting the degree to which the subject's molecular-enriched FC value for that brain region is higher or lower than the normal population ranges, respectively" – I believe this is more clear than before, but I am afraid that this would remain still obscure to someone not familiar with REACT. Can you please explain what does it mean, in biological terms, that molecular-enriched FC is higher or lower in a given subject? E.g. if higher, does it indicate that functional connectivity, in areas where a given neurotransmitter is more expressed, is more "represented" among the BOLD fluctuations of that subject? So, does this mean that functional connectivity in that subject is more strongly influenced by a given neurotransmitter system? Or rather that the neurotransmitter system is more "active" in that subject? It would be great to better explain the biological meaning of molecular-enriched FC and in which ways it can or cannot be interpreted by the reader, already in the introduction.

17) The comment about spatial resolution did not refer to the voxel size but to the possibility to distinguish two points x mm apart. Even if you resample to 2mm, your SPECT images will still have poorer resolution than your PET templates. My question is whether such differences in spatial resolution (leading to higher spatial correlation among contiguous voxels) would or not affect your results (e.g. increasing "collinearity" among your voxels). I also have some concerns about the inclusion, in the analysis, of voxels where the neurotransmitter density and hence the tracer uptake is so low that the signal there is hardly reliable/reproducible (e.g. super low signal in some areas for DAT or VACht). I'd appreciate if the authors could explain the reasoning behind that choice.

Supplementary Fig 3: what does "molecular density" mean (is it BP..?) ? how were the maps in supplementary figure 3 quantified? I'd report the actual metric used for quantification for transparency

Response to reviewers document

We thank the reviewer for further helpful comments. Here, we intersperse our response our final responses to these queries.

Reviewer #2 (Remarks to the Author):

The authors addressed most of my comments. However, there is still some clarifications that I'd appreciate seeing implemented. Numbers here refer to the number of the comments in the first revision round

1) "These deviations can be either positive or negative values reflecting the degree to which the subject's molecular-enriched FC value for that brain region is higher or lower than the normal population ranges, respectively" – I believe this is more clear than before, but I am afraid that this would remain still obscure to someone not familiar with REACT. Can you please explain what does it mean, in biological terms, that molecular-enriched FC is higher or lower in a given subject? E.g. if higher, does it indicate that functional connectivity, in areas where a given neurotransmitter is more expressed, is more "represented" among the BOLD fluctuations of that subject? So, does this mean that functional connectivity in that subject is more strongly influenced by a given neurotransmitter system? Or rather that the neurotransmitter system is more "active" in that subject? It would be great to better explain the biological meaning of molecular-enriched FC and in which ways it can or cannot be interpreted by the reader, already in the introduction.

Thank you for your comments and for emphasizing the need for clarity in explaining the biological implications of molecular-enriched FC in our REACT methodology and its implications.

In our study, the deviation scores, whether positive or negative, indicate the extent to which a subject's molecular-enriched FC for a specific brain region (ROI), deviates from the normative FC model established from a healthy control population. When we observe a positive deviation value in an ROI, this indicates that the functional coupling of this ROI with areas of high receptor expression is more pronounced than typically observed in the healthy population. This enhanced coupling suggests potentially more synchronous or integrated activity within this ROI as part of the brain's molecular-enriched network. Conversely, a negative deviation value indicates that the ROI's FC with areas of high receptor expression is reduced, implying a decrease in synchronization or integration between the ROI and the receptor-rich regions compared to the norm. Overall, these deviations can be interpreted as the network's functioning being either more or less integrated or coherent than what is established as normal among healthy subjects. Such variations can provide insights into the underlying neurobiological processes and may be associated with specific behavioural outcomes or disease states.

It is crucial to clarify that these deviations in molecular-enriched FC reflect variations in connectivity patterns and are not indicative of direct changes in neurotransmitter activity. This distinction is important as it informs the reader that while REACT provides insights into how neurotransmitter-enriched networks might function differently across individuals, it does not measure neurotransmitter activity directly.

To clarify this point, the sentence “These deviations can be either positive or negative values reflecting the degree to which the subject’s molecular-enriched FC value for that brain region is higher or lower than the normal population ranges, respectively” has been further expanded as follows: “These deviations, represented as either positive or negative values, indicate how much a subject’s molecular-enriched FC for a specific brain region deviates from the established normative FC model derived from healthy controls. A positive deviation value in a certain ROI indicates that its functional coupling with areas of high receptor density is more pronounced than typically observed, reflecting potentially more synchronous or integrated activity in this ROI within the brain’s molecular-enriched network. Conversely, negative values indicate that the ROI’s FC with areas of high receptor density is reduced, suggesting a decrease in synchronization or integration within these receptor-rich regions compared to the norm. It is crucial to note that while these deviations in molecular-enriched FC reveal variations in connectivity patterns, they do not directly measure or imply changes in neurotransmitter activity. Instead, these metrics help us understand how networks, hypothesized to be modulated by specific neurotransmitter systems based on receptor density, function in comparison to a normative framework.” We recognize that, from a biological perspective, this concept might seem less straightforward than the PET-based notion of a neurotransmitter system being more 'active'. However, we hope that our detailed explanation will help readers unfamiliar with REACT and the statistical underpinnings of functional connectivity to better understand this concept.

To explain the concept of molecular-enriched FC in the introduction, as recommended by the reviewer, we have also integrated the following part: “Specifically, molecular-enriched FC provides an indication on how each brain region, or voxel, interacts with brain areas where a certain receptor is highly distributed, providing a framework for understanding how specific neurotransmitter systems, identified by their receptor distributions, influence brain connectivity patterns. It provides a unique perspective on the functional architecture of the brain, suggesting how neurotransmitter-specific networks could modulate functional connectivity. However, it is important to note that since BOLD fMRI has no intrinsic selectivity to any neurochemical target, it does not directly measure neurotransmitter activity, nor does it imply changes in neurotransmitter levels.”

17) The comment about spatial resolution did not refer to the voxel size but to the possibility to distinguish two points x mm apart. Even if you resample to 2mm, your SPECT images will still have poorer resolution than your PET templates. My question is whether such differences in spatial resolution (leading to higher spatial correlation among contiguous voxels) would or not affect your results (e.g. increasing “collinearity” among your voxels). I also have some concerns about the inclusion, in the analysis, of voxels where the neurotransmitter density and hence

the tracer uptake is so low that the signal there is hardly reliable/reproducible (e.g. super low signal in some areas for DAT or VACht). I'd appreciate if the authors could explain the reasoning behind that choice.

Apologies for misunderstanding your original comment. Thanks for pointing this out, it is indeed a very relevant observation. We have now highlighted this within the limitations in the following sentence: "However, it is important to acknowledge these templates were acquired in separate subjects and using different methodologies, resulting differences in resolution despite being normalised into the same space".

With regards to regions with low BP (which after the normalization will have values close to 0), the fact that we employ the full spatial maps of receptor or transporter density to weight the fMRI signal means that areas with low receptor density will minimally contribute to the estimation of the dominant BOLD fluctuations within the specific network under investigation, while regions with higher density will have a greater impact. The fact that these values are low, indicating less reliability and reproducibility, means that they are considered minimally in the computation of these dominant fluctuations. By incorporating these low values rather than masking them out, we ensure that all maps can be included in a single model, vastly improving the robustness and statistical validity of our approach.

Supplementary Fig 3: what does "molecular density" mean (is it BP..?)? how were the maps in supplementary figure 3 quantified? I'd report the actual metric used for quantification for transparency

Thank you for pointing out the ambiguity of this label. It is BP for some and SUVR for others, hence we used a general term to try capture both. We have now clarified this in the figure legend as well as in the section describing the molecular templates with the following added sentence: "Molecular density reflects normalised values which are standardized uptake value ratio (SUVR: DAT and vACHT) or binding potential (BP: NAT, SERT, GABA-A, and mGluR5) within the original PET data".